# END-TO-END DOCUMENT UNDERSTANDING VIA CHAIN-OF-READING

## ABSTRACT

Intelligent Document Analysis (IDA) is a formidable task owing to documents' complex layouts, dense tables, charts, and mixed modalities. Conventional pipelines apply OCR before large language model reasoning but suffer from error propagation. End-to-end multimodal models avoid explicit pipelines yet struggle to scale to multi-page documents, where information dilution and evidence localization remain major bottlenecks. We propose Chain-of-Reading (CoR), an end-to-end framework that transforms traditional text-centric reading into a native multimodal paradigm. CoR directly consumes PDF pages as visual input, mimicking human eyes, and performs document-level question answering through a chain-of-thought process. It first localizes relevant evidence, then selectively applies OCR, and finally performs reasoning over the localized content. To further enhance comprehension of visual elements such as charts and scientific figures—which exacerbate information dilution and impede pinpointing evidence—we introduce Masked Auto-Regression (Mask-AR), a self-supervised method for multimodal grounding. CoR achieves a 14.3% improvement over the base model on the MMLongBench-Doc benchmark. We will release the CoR-Dataset and our fine-tuned model, Qwen2.5-VL-CoR.

## 1 INTRODUCTION

The proliferation of Large Language Models (LLMs) has precipitated a paradigm shift in Intelligent Document Analysis (IDA). Nonetheless, a formidable challenge persists: enabling these models to achieve deep semantic comprehension of complex, visually-rich documents, such as PDFs. These documents, curated for human readership, fuse text, charts, and intricate layouts into a semi-structured format that poses a substantial barrier to information extraction and query reasoning. The key problems in this field, therefore, converge on the imperative to develop models that can accurately and efficiently reason over information embedded within these complex visual layouts.

Two dominant paradigms address this challenge. The first relies on a pipeline-based approach, executing tasks sequentially, such as layout analysis, OCR, and specialized recognition for tables or formulas (Livathinos et al., 2025; Cui et al., 2025). Although modular, this approach suffers from high complexity and maintenance overhead. More importantly, it is highly susceptible to cascading errors: a single inaccuracy from an upstream module, like OCR, can propagate through the pipeline and compromise the integrity of the final output.

The second paradigm focuses on end-to-end solutions that bypass traditional OCR, including OCR-free Multimodal LLMs (MLLMs) (Ye et al., 2023; Wei et al., 2024) and multi-modal Retrieval-Augmented Generation (RAG) systems (Faysse et al., 2024). RAG first retrieves relevant document patches and then feeds them to a model for generation; however, decoupling retrieval from reasoning often makes the retriever a critical bottleneck. A more promising direction involves MLLMs that learn to read, localize, and reason directly from raw document pixels, integrating comprehension and reasoning within a single end-to-end framework.

Despite their potential, existing MLLMs exhibit substantial performance limitations when processing long multi-modal documents. Their effectiveness diminishes in multi-page scenarios due to two main challenges: **key information dilution** and **evidence localization difficulty** (Ma et al., 2024; Deng et al., 2024). As input sequences grow, models struggle to identify relevant passages, and they

Figure 1: Comparison of pipeline-based methods and our Chain-of-Reading (CoR) framework for document understanding

often miss critical visual cues embedded in tables or charts. These shortcomings frequently result in reasoning errors or factual hallucinations, which significantly constrain their practical utility.

To address these issues, we introduce the **"Chain-of-Reading" (CoR)**, a training paradigm inspired by human cognitive strategies for document analysis (Figure 1). CoR guides the model to first construct an explicit information-gathering path before performing complex reasoning. Under CoR, the model learns to first *locate* evidence—pinpointing relevant texts, charts, or pages—and then performs *integrated reasoning* upon this grounded foundation. This process mirrors the human cognitive pattern of scanning for key information before conducting an in-depth analysis. Furthermore, given that chart comprehension presents a distinct and formidable challenge, we also designed **Masked Auto-Regression (Mask-AR)**, an efficient self-supervised method aimed at bolstering the model's fine-grained comprehension of such complex visual elements.

Our main contributions are as follows:

- We propose **Chain-of-Reading (CoR)**, a novel training paradigm that effectively addresses evidence localization in long PDF documents and reduces hallucination.
- We introduce **Masked Auto-Regression (Mask-AR)**, a self-supervised method that substantially enhances fine-grained, multimodal comprehension of complex charts.
- We construct and release the **CoR-dataset**, the first dataset specifically designed for CoR training, curated through a low-cost, high-quality data generation pipeline.
- We develop and open-source **Qwen2.5-VL-CoR**, an end-to-end document understanding model. Experiments on long-document benchmarks demonstrate that our model achieves significant improvements, surpasses existing open-source methods—including agentic approaches—and reaches performance comparable to leading proprietary MLLMs.

## 2 RELATED WORK

### 2.1 INTELLIGENT DOCUMENT ANALYSIS

Intelligent Document Analysis (IDA) is a foundational discipline for extracting and reasoning over complex documents prevalent in fields like finance, law, and science. The contemporary landscape of IDA is largely defined by a dichotomy between pipeline-based and end-to-end methodologies.

Pipeline-based methods orchestrate a sequence of specialized modules. These systems typically commence with OCR engines or PDF parsers to extract raw text and layout information, which is then fed into a downstream LLM for semantic processing (Xie et al., 2024; Wang et al., 2024a). This modular architecture permits the integration of powerful, task-specific models for layout analysis, table recognition, and formula parsing (Huang et al., 2022; Blecher et al., 2023), as exemplified by systems like DocLayLLM and DocFormer (Liao et al., 2025; Appalaraju et al., 2021). However, this approach harbors a critical vulnerability: its susceptibility to cascading errors, where upstream inaccuracies can irrevocably degrade downstream performance.

To circumvent this fragility, end-to-end methods have emerged as a compelling alternative. These models employ a single, unified MLLM to process document images directly, thereby obviating

fragile intermediate steps. This OCR-free philosophy was pioneered by models like Donut (Kim et al., 2021) and Pix2Struct (Lee et al., 2023), which reframe document understanding as a direct image-to-sequence task. Recent advancements, such as mPLUG-DocOwl 1.5 and TextMonkey, have further enhanced cross-page understanding and robustness in text-dense scenarios (Hu et al., 2024; Liu et al., 2024). State-of-the-art models like Qwen2.5-VL now demonstrate capabilities that are closing the gap with proprietary systems like GPT-4V on a spectrum of document-centric tasks (Bai et al., 2025; Yang et al., 2023). Despite these advances, such models still grapple with the core challenges of information dilution and evidence localization in long documents—the precise gap our work aims to address.

## 2.2 Multimodal Large Models and Reasoning Strategies

The fusion of vision and language within MLLMs has unlocked new frontiers in complex reasoning. Architecturally, these models typically consist of a vision encoder, a projection layer for modality alignment, and an LLM backbone for inference. The rapid evolution of open-source models, including the InternVL series and MiniCPM-V, has been remarkable, steadily narrowing the performance chasm with their proprietary counterparts on diverse multimodal benchmarks (Chen et al., 2024c;b; Yao et al., 2024).

To elevate their reasoning capabilities from simple perception to complex cognition, strategies like Chain-of-Thought (CoT)(Wei et al., 2022) have been adapted for the multimodal domain (MCoT) (Wang et al., 2025). By generating explicit intermediate reasoning steps, MCoT enhances both model transparency and performance, a benefit substantiated by methods such as DDCoT and Compositional CoT (Mitra et al., 2024). Such explicit cognitive pathways have been shown to not only boost task performance but also to mitigate the propensity for model hallucination (Wang et al., 2025).

However, for all their success, standard CoT variants overlook a crucial step in the human cognitive process for document analysis: the distinct, sequential act of first locating relevant information before engaging in reasoning. This observation forms the central motivation for our work. While recent efforts have begun to touch upon similar concepts—for instance, SV-RAG employs an MLLM as a retriever to first select evidence (Chen et al., 2024a)—they often remain within a retrieve-then-reason paradigm rather than an integrated, trainable process. The acute challenges highlighted by benchmarks like LongDocURL and MMLongBench-Doc further underscore the urgent need for a more integrated paradigm (Deng et al., 2024; Ma et al., 2024). Drawing conceptual support from the "multi-paradigm collaboration" ideology in mathematical reasoning (Yu et al., 2025), our **Chain-of-Reading (CoR)** formalizes this "locate-then-reason" sequence into a trainable, end-to-end paradigm tailored specifically for deep document understanding.

## 3 Method

### 3.1 Chain-of-Reading

#### 3.1.1 Analysis of Failure Cases

To identify the bottlenecks of existing methods, we systematically diagnose the failure cases of leading MLLMs on long-document benchmarksA.10.1. Our analysis reveale two fundamental bottlenecks in long-document processing.

**Macroscopic Level: Contextual Inertia and Localization Failure.** Models exhibit significant "inertia" when processing long sequences, with sharp performance degradation when critical evidence is buried deep within a document or requires cross-page integration. This inefficient retrieval is a contributing factor to localization failure.

**Microscopic Level: Shallow Comprehension.** Even when evidence is successfully located, models could still fail due to insufficient depth of understanding. As shown in Figure 2, they struggle to disambiguate nuanced concepts, perform calculations on tabular data, or interpret complex charts, indicating that they merely succeed in locating evidence but fail to demonstrate genuine comprehension.

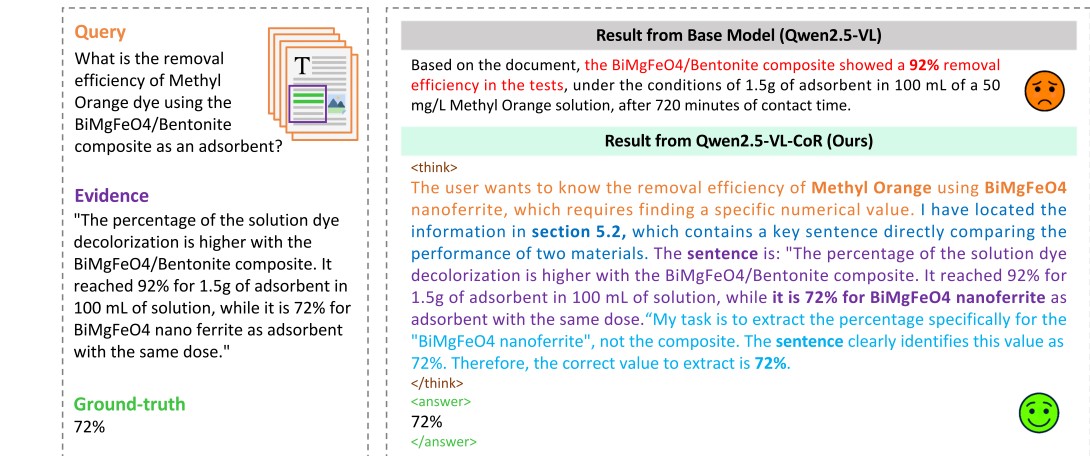

Figure 2: Qualitative example illustrating the effectiveness of Chain-of-Reading (CoR). The CoR process is segmented as follows: task planning (orange), phased & focused search (dark blue), cross-modal evidence integration (purple), and synthesized reasoning & verification (blue).

### 3.1.2 THE "CHAIN-OF-READING" PARADIGM

To address both macroscopic localization and microscopic comprehension bottlenecks, we propose the **Chain-of-Reading (CoR)** paradigm. CoR emulates an expert's reading process by transforming unstructured exploration into a structured reasoning chain, as exemplified in Figure 3. The process consists of the following four stages.

First, the **Task Planning** stage, in which the model formulates a retrieval strategy based on the query and document structure, such as prioritizing the "Methodology" section for technical questions. Second, the **Phased & Focused Search** stage, during which the model executes a coarse-to-fine iterative search, transforming the needle-in-a-haystack problem into a logical workflow of (1) scope reduction, (2) snippet localization, and (3) field extraction. Third, the **Cross-modal Evidence Integration** stage, in which the model aggregates all the textual and visual evidence located. Finally, the **Synthesized Reasoning & Verification** stage, during which the model reasons over only the integrated evidence chain to generate the answer, effectively minimizing hallucination and reducing computational overhead.

Fine-tuning on data with explicit CoR traces significantly enhances model performance in long-document scenarios. For a comprehensive gallery of detailed, step-by-step examples illustrating CoR in various complex scenarios, please see the Appendix A.9.

### 3.2 MASKED AUTO-REGRESSION FOR VISUAL GROUNDING

While CoR establishes a macroscopic reasoning framework, achieving a deep understanding of information-dense and complex visual elements like charts and diagrams requires finer-grained perception and reasoning abilities. However, constructing large-scale supervised datasets for such fine-grained tasks is often prohibitively expensive. To address this challenge, we introduce **Masked Auto-Regression (Mask-AR)**, an efficient self-supervised learning method designed to enhance the model's capacity for in-depth analysis of complex visual artifacts.

In this method, we use scientific literature as the primary data source. Scientific articles contain abundant figures and tables, whose captions or titles provide dense, expert-authored summaries of the most important multimodal information. We design a pre-training task in which the MLLM model receives native multimodal inputs (PDF page image sequences), while a layout model masks portions of key caption text. The Mask-AR task is formulated to reconstruct masked text segments by leveraging both the visual content and the surrounding contextual information.

This task encourages the model to form deep cross-modal associations between visual features and textual semantics. The self-supervised process implicitly guides the model to analyze the internal

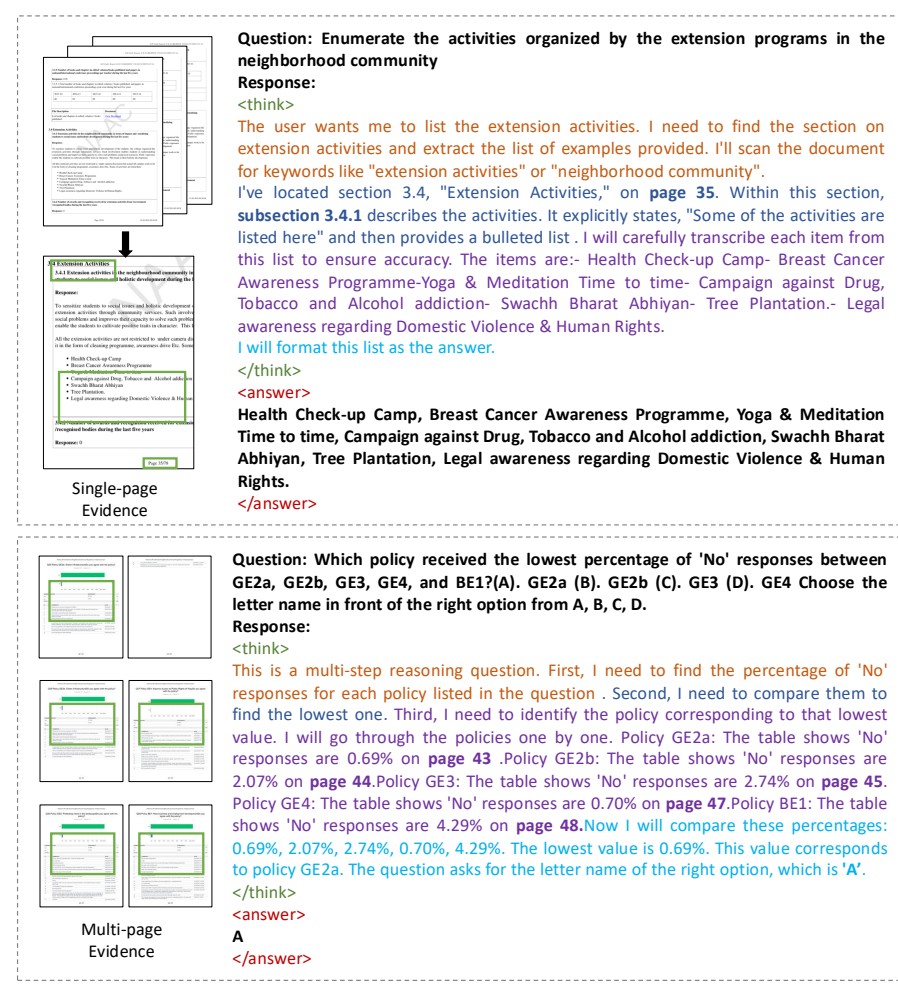

Figure 3: Exemplars of the Chain-of-Reading (CoR) paradigm in action. Top: precise localization and extraction of a list from a specific section; Bottom: cross-page evidence retrieval and comparison to identify the minimum value across multiple tables.

structure of visual elements—such as complex model architectures or multi-step flowcharts—and accurately align these visual cues with their corresponding textual descriptions.

This enhanced comprehension is crucial for complex tasks, such as identifying and rejecting questions based on false premises, as demonstrated in Appendix A.9.11, Example 11. Implementation details are provided in Appendix A.1.

By leveraging abundant figure-caption pairs in scientific documents, Mask-AR offers a fully self-supervised, data-efficient, and scalable approach for developing advanced visual reasoning capabilities.

## 4 DATASET AND TRAINING

### 4.1 DATASET CONSTRUCTION

#### 4.1.1 MOTIVATION AND THE CoR-DATASET

The advancement of long-document understanding has been critically hindered by the scarcity of appropriate training data. Most existing VQA and document analysis datasets are confined to single-page input (Huang et al., 2022; Masry et al., 2022), a limitation that precludes models from de-

veloping the cross-page reasoning and evidence aggregation capabilities essential for real-world applications involving multi-page reports or scholarly articles.

To address this critical deficit, we construct the **CoR-Dataset**, a resource specifically engineered following our Chain-of-Reading paradigm. The dataset was curated using a novel, low-cost semi-automated pipeline that yields high-fidelity data, as depicted in Figure 4. This process integrates guided data generation with automated quality assessment and iterative refinement, ultimately yielding 26 088 high-quality QA pairs. Each pair is annotated with an explicit reasoning trace that materializes the structured "reading chain," providing the direct supervision necessary for our training approach. **A detailed statistical breakdown of the CoR-Dataset's composition, including distributions of document types, question intents and reasoning complexity, is provided in Appendix A.8.** A detailed breakdown of each stage in our data generation pipeline is provided in Appendix A.2.

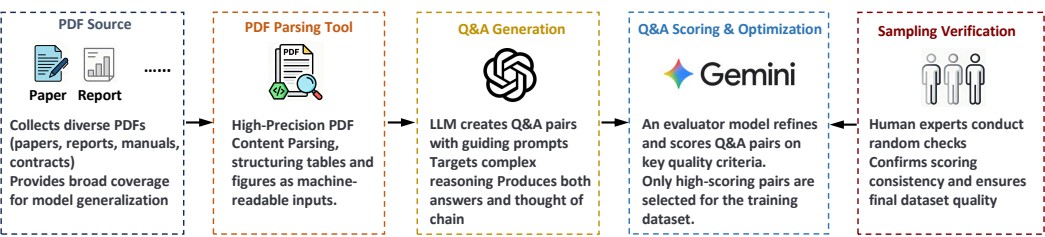

Figure 4: Overview of our data generation pipeline for the CoR-Dataset. The process involves document parsing, guided Q&A generation with reasoning trace annotation, automated scoring and refinement, and final human verification. Full details are in Appendix A.2.

### 4.1.2 MASK-AR DATASET

The training data for our Mask-AR objective are also sourced from our extensive corpus of scientific documents. We note that naive extraction of all figure-caption pairs yields a dataset fraught with low-quality and irrelevant samples (e.g., simple logos or decorative images). To ensure that the self-supervised task is both challenging and semantically meaningful, we engineer a sophisticated filtering pipeline, as depicted in Figure 5. Following an initial PDF parsing with Uni-Parser(Team, 2025), a high precision PDF parsing framework, we employ a powerful MLLM (Gemini-2.5-Pro), which acts as an expert surrogate to programmatically identify and select the most valuable samples. This curation process is guided by criteria that prioritize pairs exhibiting high information density in the caption and substantial visual complexity in the figure, such as architectural diagrams or plots of experimental results. This meticulous curation is indispensable for creating a dataset that guides the model to develop deep visual-textual reasoning skills. The complete step-by-step methodology is further detailed in Appendix A.1.

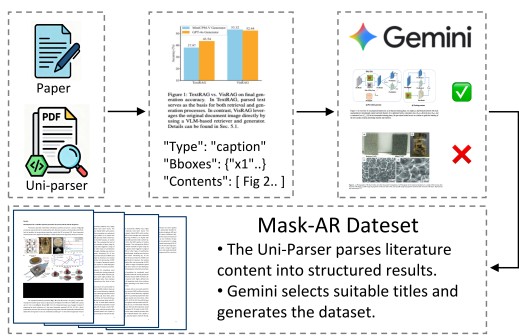

Figure 5: Data curation pipeline for the Mask-AR self-supervised task. Uni-Parser extracts figure-caption pairs, after which a powerful MLLM (Gemini-2.5-Pro) acts as an expert surrogate to filter for high-quality, information-dense samples, ensuring the effectiveness of the training set.

### 4.2 THREE-STAGE TRAINING STRATEGY

Our training recipe is progressive, in a three-stage framework, designed methodically to comprehensively enhance the model capabilities of document analysis.

**Stage 1: Foundational Capability Enhancement.** We start by bolstering the foundational capabilities of the base model (Qwen2.5-VL-7B). Using Low-Rank Adaptation (LoRA) Hu et al. (2022), we perform parameter-efficient fine-tuning on a curated mixture of publicly available document analysis datasets. This foundational training is designed to enhance the model's core competencies in visual text recognition, layout understanding, and table/chart parsing. **A comprehensive list of the datasets employed is detailed in Appendix A.7.** This stage focuses updates on the language model components while the visual encoder remained frozen.

**Stage 2: Task-Specific Fine-tuning.** The model then undergoes full-parameter fine-tuning on the language model components using our proprietary **CoR-Dataset** and **Mask-AR dataset**. This crucial stage deeply ingrains the CoR reasoning patterns and enhances its visual grounding abilities. The training is specifically structured to remediate common failure modes identified in our analysis, such as evidence hallucination, format inconsistency, and superficial content retrieval.

**Stage 3: Preference Alignment.** In the final stage, we align the model's outputs with human preferences for quality, reliability, and helpfulness using Direct Preference Optimization (DPO) Rafailov et al. (2023). We train the model on a custom-built preference dataset of 5,000 pairs. The preferred (chosen) responses are high-quality examples from our CoR-Dataset, while the undesirable (rejected) responses are synthetically generated to reflect the common error patterns identified. To enhance training stability and mitigate the impact of potential label noise, we employ a hybrid loss function combining the standard sigmoid loss with a robust variant. The mathematical formulation and further details are available in Appendix A.3.

## 5 EXPERIMENTS

### 5.1 EXPERIMENTAL SETUP

We fine-tune the Qwen2.5-VL-7B model, henceforth referred to as **Qwen2.5-VL-CoR-7B**. We conduct a comprehensive evaluation, benchmarking our model against its base version as well as series of leading open-source and proprietary models. Detailed training configurations are provided in Appendix A.5.

**Evaluation Benchmarks.** We evaluate model performance on two challenging public benchmarks for long-document multimodal question answering: **MMLongBench-Doc** Ma et al. (2024) and **LongDocURL** Deng et al. (2024). These benchmarks are specifically selected as they feature lengthy, multi-page documents and complex queries that necessitate synthesizing evidence across multiple pages and modalities. Consequently, they serve as an ideal testbed for evaluating the core capabilities our work aims to enhance. A detailed statistical breakdown of each benchmark is available in Appendix A.4.

**Evaluation Metrics.** For both benchmarks, we strictly adhere to their official evaluation protocols. To facilitate a granular analysis, we report accuracy disaggregated by both the modality of the required evidence and the number of pages from which information must be synthesized. Furthermore, we report the overall **generalized accuracy** and **F1 score** to provide a holistic view of performance. The main results are presented in Table 1 and Table 2.

### 5.2 MAIN RESULTS AND ANALYSIS

The experimental results, presented in Table 1 and Table 2, unequivocally demonstrate the substantial performance gains conferred by our proposed framework.

**Dominant Performance on MM-LongBench-Doc.** As shown in Table 1, Qwen2.5-VL-CoR-7B establishes a new state-of-the-art among open-source end-to-end models. It achieves an overall accuracy of **37.4%** and an F1-score of **36.0%**, decisively outperforming its base model (23.1% Acc) by a remarkable margin of +14.3 percentage points. This substantial delta underscores the profound impact of our CoR training paradigm and multi-stage fine-tuning strategy.

In a broader comparison, Qwen2.5-VL-CoR-7B not only surpasses all open-source rivals like Docopilot-8B but also outperforms formidable proprietary models such as GPT-4V (32.4%). While

Table 1: Detailed performance on the **MM-LongBench-Doc** benchmark. The **best overall** score in each column is bolded, and the best open-source score is underlined. [†]Results are from Han et al. (2025), Duan et al. (2025), or the official benchmark paper (Ma et al., 2024). Abbreviations: SIN (single-page), MUL (multi-page), UNA (unanswerable). All scores are in percentage (%).

| Method | Overall | | By Page Count (Acc.) | | | By Evidence Source (Acc.) | | | | |
|---|---|---|---|---|---|---|---|---|---|---|
| | ACC | F1 | SIN | MUL | UNA | TXT | CHA | LAY | TAB | FIG |
| *Non-End-to-End Methods (RAG, etc.)* | | | | | | | | | | |
| OCR(Tesseract)+GPT-4o[†] | 30.5 | 30.1 | 35.4 | 29.3 | 18.6 | 41.1 | 23.4 | 28.5 | 38.1 | 22.4 |
| MDocAgent[†] | 31.5 | — | — | — | — | 34.7 | 32.3 | 40.1 | 29.4 | 32.1 |
| *End-to-End Methods (Open-source)* | | | | | | | | | | |
| Docopilot-8B[†] | 28.8 | 23.0 | — | — | — | — | — | — | — | — |
| Qwen2.5-VL-7B | 23.1 | 22.5 | 24.3 | 16.5 | 31.1 | 27.4 | 20.5 | 25.2 | 22.4 | 20.3 |
| **Qwen2.5-VL-CoR-7B (Ours)** | 37.4 (+14.3) | 36.0 (+13.5) | 41.9 | 25.9 | 45.5 | 39.4 | 27.7 | 31.2 | 38.6 | 27.5 |
| *End-to-End Methods (Proprietary)* | | | | | | | | | | |
| GPT-4V[†] | 32.4 | 31.2 | 36.4 | 27.0 | 31.2 | 34.4 | 28.3 | 28.2 | 32.4 | 26.8 |
| Gemini-1.5-Pro[†] | 28.2 | 20.6 | 21.1 | 11.1 | **69.2** | 21.0 | 17.6 | 6.9 | 14.5 | 15.2 |
| GPT-4o[†] | **42.8** | **44.9** | **54.5** | **41.5** | 20.2 | **46.3** | **46.0** | **45.3** | **50.0** | **44.1** |

Table 2: Detailed performance on the **LongDoc-URL** benchmark. The **best overall** score is bolded, and the best open-source score is underlined. [†]Results are from Han et al. (2025) or the official benchmark paper (Deng et al., 2024). All scores are reported as Accuracy (%).

| Method | Overall | Main Task | | | Element Type | | | | Evidence Pages | |
|---|---|---|---|---|---|---|---|---|---|---|
| | ACC | UND | REA | LOC | TXT | LAY | FIG | TAB | SIN | MUL |
| *Non-End-to-End Methods (Agent-based, etc.)* | | | | | | | | | | |
| OCR(PyMuPDF) + GPT-4o[†] | 34.7 | 35.3 | 28.0 | 37.2 | 34.3 | 33.7 | 35.0 | 26.9 | 28.2 | 35.1 |
| OCR(PyMuPDF) + o1-preview[†] | 35.8 | 35.6 | 30.6 | 38.6 | 33.2 | 36.8 | 35.9 | 33.0 | 29.1 | 37.1 |
| MDocAgent[†] | 51.7 | — | — | — | — | — | — | — | — | — |
| *End-to-End Methods (Open-source)* | | | | | | | | | | |
| Qwen2-VL-7B[†] | 30.6 | 36.8 | 24.0 | 22.6 | 33.4 | 38.2 | 30.9 | 24.3 | 26.4 | 34.4 |
| Qwen2.5-VL-7B | 39.2 | 44.5 | 31.2 | 33.5 | 42.8 | 43.9 | 37.5 | 33.3 | 36.5 | 41.0 |
| **Qwen2.5-VL-CoR-7B (Ours)** | 51.5 (+12.3) | 56.3 | 41.2 | 48.6 | 55.6 | 51.4 | 48.2 | 46.2 | 51.8 | 51.3 |
| *End-to-End Methods (Proprietary)* | | | | | | | | | | |
| Qwen-VL-Max[†] | 49.5 | 58.9 | 43.9 | 36.0 | 53.5 | 55.2 | 52.5 | 46.7 | 50.9 | 51.9 |
| Gemini-1.5-Pro[†] | 50.9 | 55.6 | 42.3 | 46.4 | 51.8 | **56.1** | 52.1 | 43.1 | 44.4 | 53.5 |
| GPT-4o[†] | **64.5** | **68.6** | **59.3** | **59.6** | **66.3** | **64.1** | **67.5** | **60.2** | **62.2** | **65.7** |

the latest GPT-4o model sets a high ceiling at 42.8%, our 7B-parameter model exhibits highly competitive performance. The disaggregated results further illuminate our model's strengths, revealing significant gains in both single-page (SIN) and, critically, multi-page (MUL) reasoning scenarios, alongside robust improvements across all evidence modalities.

**Leading Performance on the More Demanding LongDoc-URL Benchmark.** The LongDoc-URL benchmark, characterized by its significantly longer documents, poses a more formidable challenge to long-context reasoning. On this rigorous testbed (Table 2), Qwen2.5-VL-CoR-7B continues its exceptional performance, achieving an overall accuracy of **51.5%**. This result cements our model as the **premier open-source end-to-end solution**, again showcasing a massive improvement of +12.3 points over its base model.

Crucially, our model's performance transcends the open-source sphere and is highly competitive with top-tier proprietary systems. It is particularly noteworthy that Qwen2.5-VL-CoR-7B (51.5%) effectively matches the performance of the powerful, agent-based MDocAgent system (51.7%) and **surpasses other leading proprietary models, including Qwen-VL-Max (49.5%) and Gemini-**

**1.5-Pro (50.9%)**. This is a remarkable achievement for a 7B-parameter model, demonstrating that our targeted training approach can bridge the performance gap typically attributed to massive model scale or complex external tool usage. The ability to outperform larger proprietary models underscores the efficiency and power of instilling structured reasoning directly into the model.

**Summary of Experimental Findings.** In summary, our comprehensive evaluations on two demanding long-document benchmarks validate the superiority of our methodology. Qwen2.5-VL-CoR-7B consistently sets a new standard for open-source models in this domain. The results furnish compelling evidence that with a principled, data-centric approach to teaching structured reasoning, smaller models can not only compete with but, in certain cases, surpass their much larger, proprietary counterparts.

## 5.3 ABLATION STUDIES

To rigorously dissect the contribution of each component within our framework, we conduct a comprehensive ablation study. We systematically evaluate the incremental impact of Supervised Fine-Tuning (SFT) on our Chain-of-Reading (CoR) and Mask-AR datasets, followed by Direct Preference Optimization (DPO). The results are summarized in Table 3.

Table 3: Main ablation study on overall accuracy (%). The checkmarks (✓) indicate which components are included in each configuration. The performance gains for each step are shown relative to the base model.

| Configuration | Components | | | Overall Accuracy (%) | |
|---|---|---|---|---|---|
| | CoR SFT | Mask-AR | DPO | MMLongBench | LongDocURL |
| Base Model | | | | 23.1 | 39.2 |
| + CoR | ✓ | | | 34.0 **(+10.9)** | 47.0 **(+7.8)** |
| + CoR + Mask-AR | ✓ | ✓ | | 35.1 **(+12.0)** | 48.1 **(+8.9)** |
| + CoR + DPO | ✓ | | ✓ | 35.9 **(+12.8)** | 48.9 **(+9.7)** |
| **Full Model (Ours)** | ✓ | ✓ | ✓ | **37.4 (+14.3)** | **51.5 (+12.3)** |

**Analysis of Component Synergy.** The main ablation results in Table 3 clearly elucidate the effectiveness of our multi-stage architecture. SFT with the CoR dataset provides a foundational performance boost(+10.9% and +7.8% on the two benchmarks, respectively), establishing robust reasoning capabilities . Both Mask-AR and DPO contribute further gains on top of this foundation. Critically, the full model (Row 5), which integrates all three components, achieves the highest scores, confirming a powerful synergistic effect. This indicates that enhancing visual grounding (Mask-AR) and aligning with human preferences (DPO) are complementary, rather than redundant, to the core reasoning patterns instilled by CoR.

**Component-Specific Contributions.** To further investigate these effects, we analyzed the specific roles of Mask-AR and DPO. Our fine-grained analysis reveals that Mask-AR provides a targeted boost to visual-centric questions, measurably improving accuracy on queries requiring chart and figure interpretation. Concurrently, DPO proves instrumental in refining higher-level cognitive abilities, yielding the most substantial gains in complex, multi-page reasoning tasks where nuanced judgment is paramount. A detailed breakdown substantiating these claims is provided in Appendix A.6.

## 6 CONCLUSION

This paper presents Chain-of-Reading (CoR), an end-to-end paradigm for document understanding. CoR enhances multimodal document QA by structuring document-level reasoning through explicit reasoning paths. It further leverages Masked Auto-Regression for fine-grained visual comprehension with self-supervised visual grounding. Qwen2.5-VL-CoR-7B achieves accuracy improvements of 14.3% on MMLongBench-Doc and 12.3% on LongDocURL compared to Qwen2.5-VL-7B, and, despite having only 7B parameters, delivers performance comparable to proprietary MLLMs such as GPT-4o.

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

# A APPENDIX

This appendix provides supplementary details on our methodology and dataset construction to facilitate reproducibility and deeper understanding.

## A.1 IMPLEMENTATION DETAILS OF MASKED AUTO-REGRESSION (MASK-AR)

The implementation of our Mask-AR self-supervised objective follows a structured process designed to maximize its learning signal for deep, cross-modal reasoning. The process, illustrated in the main text in Figure 5, consists of the following steps:

1. **Extraction:** We use a high-fidelity document parser (Uni-Parser) to extract all figure images and their corresponding caption texts from a large corpus of scientific and technical documents. Each figure-caption pair is maintained with a link to its source document.

2. **Intelligent Filtering:** To create a challenging and high-quality training set, we filter the extracted pairs. Each pair, along with its full document context, is evaluated by a powerful MLLM (Gemini-2.5-Pro) based on predefined criteria:
   - **Information Density:** Captions that are rich in technical details, experimental results, or key conclusions are preferred over simple descriptive labels (e.g., "Figure 1: System Overview").
   - **Visual Complexity:** Figures with multiple components, data series, complex layouts, or abstract concepts are prioritized.
   - **Content Relevance:** We select figures that are central to the document's main contributions, such as model architecture diagrams or plots of primary experimental results.

3. **Sample Construction:** For each selected document, we adhere to a "one instance per document" principle. We mask the caption of only the single most representative figure identified during the filtering stage. The training sample then consists of all pages of the document (with the target caption text masked out) and the target figure image.

4. **Training Objective:** The model is trained to auto-regressively generate the original, unmasked caption text. This task compels the model to synthesize information from both the visual data in the figure and the textual context scattered throughout the document, effectively teaching it to perform the complex cognitive process of summarizing visual evidence in context.

## A.2 DATASET CONSTRUCTION DETAILS

### A.2.1 CoR-DATASET GENERATION PIPELINE

The CoR-Dataset was constructed using the semi-automated pipeline shown in Figure 4. The four key stages are:

1. **Document Collection and Parsing:** We first gathered a diverse collection of PDF documents spanning scientific literature, financial reports, technical manuals, and legal contracts. Each document was processed with Uni-Parser, a high-performance tool that performs OCR and structures content like tables and lists, providing a clean, machine-readable foundation.

2. **Guided Q&A and CoR Generation:** The parsed document content was fed to a powerful teacher model (GPT-4o). We used carefully engineered prompts to guide the model to generate question-answer pairs that necessitate complex reasoning (e.g., cross-page comparison, chart interpretation with text). Crucially, we also prompted the model to output a detailed, step-by-step "reading chain" that explicitly follows our CoR paradigm, serving as the ground-truth reasoning path.

3. **Automated Quality Assessment and Refinement:** To ensure data quality, we employed an independent evaluator model (Gemini-2.5-Pro) to score each generated sample. The scoring criteria included the logical soundness of the question, the clarity of the CoR chain, and the factual accuracy of the answer. Low-scoring samples were either discarded or sent back to the teacher model with feedback for revision, creating a closed-loop optimization process that continuously improved data quality.

4. **Human Verification:** The final stage involved manual review and verification by human annotators to filter out any remaining subtle errors and ensure the dataset's overall reliability.

### A.3 DETAILS OF THE DPO TRAINING STAGE

In Stage 3 of our training, we used Direct Preference Optimization (DPO) to align the model with human preferences.

**DPO Loss Function.** DPO directly optimizes the policy on a dataset of ranked preferences. Given a prompt $x$ and a pair of responses $(y_w, y_l)$, where $y_w$ is the preferred (winning) response and $y_l$ is the dispreferred (losing) response, the DPO loss function is defined as:

$$\mathcal{L}_{\text{DPO}}(\pi_\theta; \pi_{\text{ref}}) = -\mathbb{E}_{(x,y_w,y_l)\sim\mathcal{D}} \left[ \log \sigma \left( \beta \log \frac{\pi_\theta(y_w|x)}{\pi_{\text{ref}}(y_w|x)} - \beta \log \frac{\pi_\theta(y_l|x)}{\pi_{\text{ref}}(y_l|x)} \right) \right] \tag{1}$$

where $\mathcal{D}$ is the preference dataset, $\pi_\theta$ is the policy model being optimized, $\pi_{\text{ref}}$ is a fixed reference model (initialized from the Stage 2 checkpoint), $\beta$ is a temperature hyperparameter, and $\sigma$ is the logistic sigmoid function.

**Preference Dataset Construction.** We constructed a high-quality preference dataset containing 5,000 pairs. The generation process was as follows:

- **Preferred Responses** ($y_w$)**:** We selected high-scoring, correct examples from a held-out portion of our CoR-Dataset. These represent the ideal model outputs in terms of format, reasoning, and accuracy.

- **Dispreferred Responses** ($y_l$)**:** We first conducted a thorough error analysis of the outputs from the Stage 2 model. Based on a typology of common errors (e.g., factual inaccuracies, evidence misattribution, format violations, lazy retrieval), we prompted Gemini-2.5-Pro to generate corresponding dispreferred responses for each prompt $x$ and its preferred response $y_w$. This ensures that the model learns to avoid specific, realistic failure modes.

**Hybrid Loss Function.** To enhance training stability and robustness against potential label noise in our synthetically-aided preference dataset, we employed a hybrid loss function that combines two variants. The total loss $\mathcal{L}_{\text{total}}$ is a weighted sum:

$$\mathcal{L}_{\text{total}} = w_1 \cdot \mathcal{L}_{\text{sigmoid}} + w_2 \cdot \mathcal{L}_{\text{robust}}, \tag{2}$$

where $w_1 = 0.7$ and $w_2 = 0.3$ (configured via `--loss_type sigmoid robust`). The components are:

- **Sigmoid Loss ($\mathcal{L}_{\text{sigmoid}}$):** This is the standard loss from the original DPO paper Rafailov et al. (2023), equivalent to Equation 1:

$$\mathcal{L}_{\text{sigmoid}}(\pi_\theta; \pi_{\text{ref}}) = -\mathbb{E}_{(x,y_w,y_l)\sim\mathcal{D}} \left[ \log \sigma \left( \beta \log \frac{\pi_\theta(y_w|x)}{\pi_{\text{ref}}(y_w|x)} - \beta \log \frac{\pi_\theta(y_l|x)}{\pi_{\text{ref}}(y_l|x)} \right) \right], \quad (3)$$

where $\sigma$ is the sigmoid function, fitting a Bradley-Terry model to the preferences.

- **Robust Loss ($\mathcal{L}_{\text{robust}}$):** This variant is an unbiased estimator of the DPO loss that is resilient to preference noise in the data Wang et al. (2024b); Chowdhury et al. (2024). It models the possibility of incorrect preference labels via a label smoothing hyperparameter $\varepsilon \in (0, 1/2)$ (the flip rate of preference labels). The loss is defined as:

$$\mathcal{L}_{\text{robust}}(\pi_\theta; \pi_{\text{ref}}) = \frac{1}{N} \sum_{i=1}^{N} \frac{(1-\varepsilon)\mathcal{L}_{\text{sigmoid}}(\pi_\theta; \pi_{\text{ref}}, x_i, \tilde{y}_{w,i}, \tilde{y}_{l,i}) - \varepsilon\mathcal{L}_{\text{sigmoid}}(\pi_\theta; \pi_{\text{ref}}, x_i, \tilde{y}_{l,i}, \tilde{y}_{w,i})}{1 - 2\varepsilon},$$
$$(4)$$

where $\tilde{y}_{w,i}$ and $\tilde{y}_{l,i}$ are the potentially noisy preferred and dispreferred responses for prompt $x_i$, and $N$ is the batch size. When $\varepsilon = 0$, this reduces to the standard sigmoid loss. In our experiments, we used $\varepsilon = 0.1$ (or specify your value if different).

## A.4 Evaluation Benchmarks

Our experiments were conducted on the following standard long-document VQA benchmarks, which are designed to test a model's ability to comprehend and reason over lengthy, visually complex documents.

- **MMLongBench-Doc** Ma et al. (2024): This benchmark consists of 135 long-form PDF documents, with an average of 47.5 pages and 21,214 tokens per document. It contains 1,082 expert-annotated questions designed to test long-context understanding.
- **LongDocURL** Deng et al. (2024): This dataset is constructed from 396 lengthy PDF documents, averaging 85.6 pages and 43,622.6 tokens. It includes 2,325 high-quality question-answering pairs. A key challenge of this benchmark is that correct answers often require synthesizing evidence from multiple modalities (e.g., text, tables, images) and across different pages.

## A.5 Training Configurations

All fine-tuning was performed on a server equipped with **8 NVIDIA A100 80GB GPUs**. The training utilized the PyTorch framework, along with libraries such as Hugging Face Transformers and Swift. The base model for all stages is **Qwen2.5-VL-7B**. Below are the specific configurations for each of our three training stages.

### A.5.1 Stage 1: Foundational Capability Enhancement (LoRA)

In this stage, we performed parameter-efficient fine-tuning using Low-Rank Adaptation (LoRA) to enhance the model's core document understanding abilities on a mixture of public datasets.

- **Method:** Low-Rank Adaptation (LoRA).
- **Trained Components:** LoRA adapters were applied to the language model's attention (`q_proj`, `k_proj`, `v_proj`, `o_proj`) and MLP (`gate_proj`, `up_proj`, `down_proj`) layers, as well as the multimodal projector (`mm_projector`). The visual encoder weights remained frozen.
- **LoRA Hyperparameters:**
  - LoRA Rank ($r$): 16
  - LoRA Alpha ($\alpha$): 32
  - LoRA Dropout: 0.05
- **Training Hyperparameters:**
  - Optimizer: AdamW
  - Learning Rate: $1.0 \times 10^{-4}$
  - LR Scheduler: Cosine decay with a 10% warmup ratio

- Global Batch Size: 64 (1 per device $\times$ 8 accumulation steps $\times$ 8 GPUs)
- Number of Epochs: 3.0
- Precision: bfloat16
- Max Sequence Length: 32,768
- Attention Implementation: Flash Attention 2
- Weight Decay: 0.05
- Gradient Clipping Norm: 0.3

### A.5.2 STAGE 2: TASK-SPECIFIC FINE-TUNING (FULL-PARAMETER)

This stage involved full-parameter fine-tuning on our proprietary CoR-Dataset and Mask-AR dataset to instill the Chain-of-Reading reasoning patterns.

- **Method:** Full-parameter supervised fine-tuning.
- **Trained Components:** We updated the full weights of the language model and the multimodal projector. The visual encoder (`vision_tower`) remained frozen throughout this stage.
- **Training Hyperparameters:**
  - Optimizer: AdamW
  - Learning Rate: $1.0 \times 10^{-5}$
  - LR Scheduler: Cosine decay with a 5% warmup ratio
  - Global Batch Size: 16 (1 per device $\times$ 2 accumulation steps $\times$ 8 GPUs)
  - Number of Epochs: 1.0
  - Precision: bfloat16
  - Max Sequence Length: 32,768
  - Parallelism Strategy: DeepSpeed ZeRO Stage 3
  - Attention Implementation: Flash Attention 2

### A.5.3 STAGE 3: PREFERENCE ALIGNMENT (DPO WITH LoRA)

In the final stage, we aligned the model with human preferences using Direct Preference Optimization (DPO). For computational efficiency, this stage was also conducted using LoRA.

- **Method:** Direct Preference Optimization (DPO) with LoRA.
- **Reference Model:** The reference model ($p_{\text{ref}}$) for calculating the KL-divergence was the checkpoint obtained at the end of Stage 2.
- **Hybrid Loss Function:** As mentioned in Section 4.2, we employed a hybrid loss function. The final loss was a weighted sum of the standard sigmoid loss and a robust loss variant: $L_{\text{hybrid}} = 0.7 \times L_{\text{sigmoid}} + 0.3 \times L_{\text{robust}}$.
- **LoRA Hyperparameters:**
  - LoRA Rank ($r$): 8
  - LoRA Alpha ($\alpha$): 32
  - Target Modules: All linear layers in the language model.
- **Training Hyperparameters:**
  - Optimizer: AdamW
  - Learning Rate: $5.0 \times 10^{-6}$
  - LR Scheduler: Cosine decay with a 5% warmup ratio
  - Global Batch Size: 16 (1 per device $\times$ 2 accumulation steps $\times$ 8 GPUs)
  - Number of Epochs: 1.0
  - Precision: bfloat16
  - Max Sequence Length: 32,767
  - Parallelism Strategy: DeepSpeed ZeRO Stage 3

### A.6 DETAILED BREAKDOWN OF ABLATION COMPONENT EFFECTS

To further dissect the results of our main ablation study (Table 3), we analyzed the specific impact of the Mask-AR and DPO stages on relevant sub-tasks.

**Effect of Mask-AR on Visual Element Understanding.** To specifically isolate the impact of the Mask-AR dataset on visual parsing, we compare performance on visually-intensive evidence types before and after its inclusion, across both benchmarks. As shown in Table 4, adding Mask-AR SFT consistently improves accuracy on questions related to figures and charts/tables. On MMLongBench-Doc, chart-related accuracy increases by **+3.1%**, while on LongDocURL, figure accuracy improves by **+3.2%**. This consistently positive impact across different benchmarks and visual types directly validates our hypothesis that Mask-AR enhances the model's ability to interpret and extract information from complex visual elements.

Table 4: Impact of Mask-AR on visual categories (Accuracy, %) across both benchmarks.

| Benchmark | Evidence Type | + CoR SFT | + CoR + Mask-AR SFT |
|---|---|---|---|
| MMLongBench | Chart (CHA) | 23.1 | **26.2** |
| | Figure (FIG) | 20.7 | **21.3** |
| LongDocURL | Figure | 44.3 | **47.5** |
| | Table | 41.8 | **42.3** |

**Effect of DPO on Higher-Level Cognitive Abilities.** We hypothesize that DPO's primary role is to refine the model's high-level cognitive abilities. To verify this, we measured its impact on complex reasoning and comprehension sub-tasks in both benchmarks. Table 5 shows that applying DPO yields significant gains in these crucial areas. It boosts multi-page reasoning on MMLongBench by a remarkable **+7.6%**, demonstrating an improved ability to synthesize information across long contexts. Similarly, on LongDocURL, it enhances both Understanding (**+3.2%**) and Reasoning (**+1.6%**). This robust evidence across two benchmarks confirms that DPO is crucial for aligning the model with nuanced human expectations, fundamentally improving its ability to think and reason through complex problems.

Table 5: Impact of DPO on reasoning and comprehension (Accuracy, %) across both benchmarks.

| Benchmark | Sub-task | SFT Only (CoR+Mask-AR) | + DPO (Full Model) |
|---|---|---|---|
| MMLongBench | Multi-page (MUL) | 18.3 | **25.9** |
| LongDocURL | Understanding (UND) | 53.1 | **56.3** |
| | Reasoning (REA) | 39.6 | **41.2** |

### A.7 DATASETS FOR STAGE 1 FOUNDATIONAL FINE-TUNING

In the first stage of our training, we performed LoRA-based fine-tuning on a diverse collection of public and curated datasets to enhance the model's fundamental document understanding skills. The datasets were carefully selected to cover a wide range of tasks, including document-based visual question answering (DocVQA), table question answering (TableQA), and chart question answering (ChartQA). This mixed-data approach ensures the model develops robust capabilities across various document types and formats before undergoing specialized training in Stage 2. Table 6 provides a detailed summary of each dataset component.

### A.8 STATISTICAL ANALYSIS OF THE CoR-DATASET

The CoR-Dataset was meticulously designed to encompass a wide diversity of documents, question types, and reasoning challenges, reflecting the complexity of real-world document analysis tasks. In total, the dataset comprises 26 087 high-quality, annotated question-answer pairs. To ensure its breadth and depth, we analyzed its composition across several key dimensions. A summary of the primary statistics is presented in Table 7, while the detailed distributions for each dimension are illustrated in **Figures 6 through 9**.

The distributions highlight a focus on academic and technical documents, which provide fertile ground for complex questions. The question intents are predominantly geared towards factual extraction, but with significant representation from summarization, comparison, and causal inquiries,

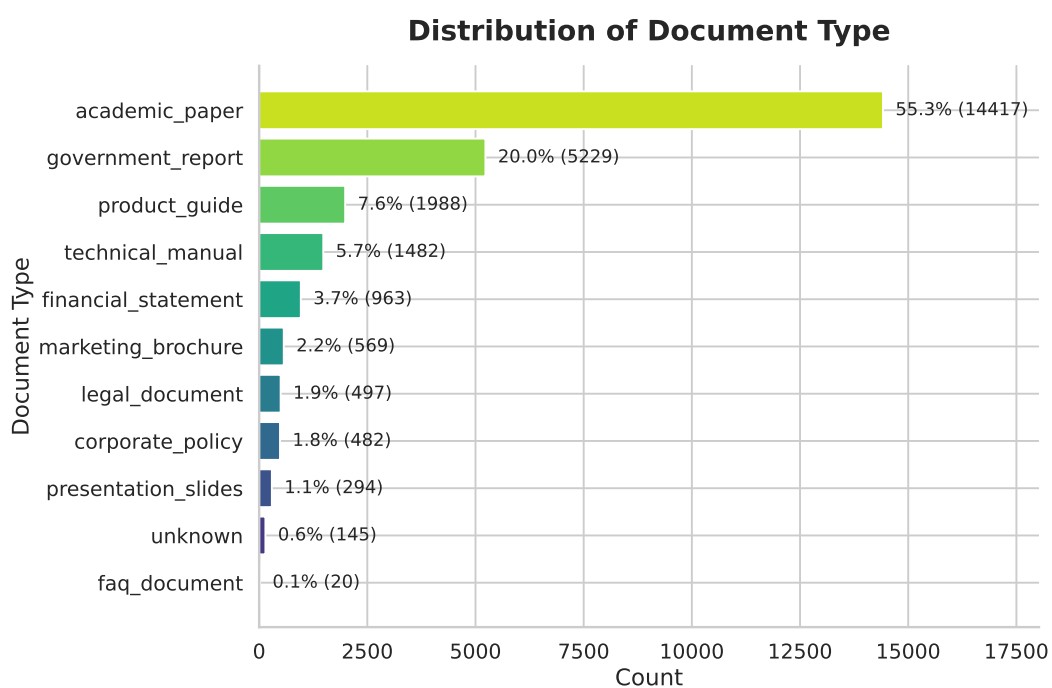

Figure 6: Distribution of document types in the CoR-Dataset. The dataset is predominantly composed of academic papers (55.3%) and government reports (20.0%), providing a rich source of structured, information-dense content for training complex reasoning.

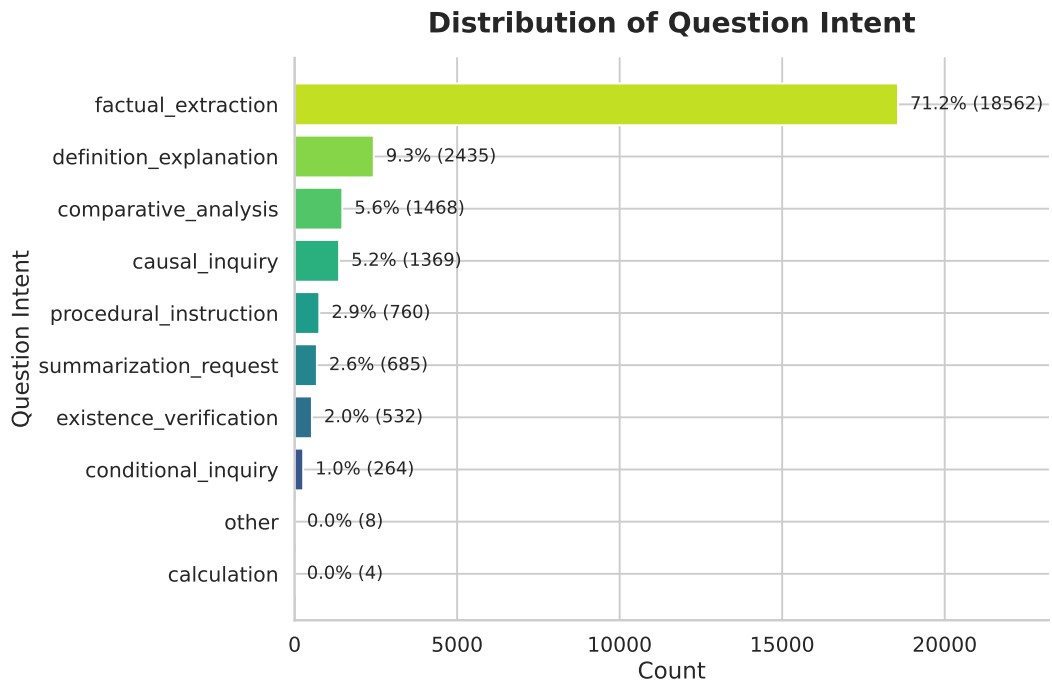

Figure 7: Distribution of question intents. While factual extraction (71.2%) forms the core, the dataset includes a significant proportion of questions requiring higher-level understanding, such as definition/explanation (9.3%) and comparative analysis (5.6%).

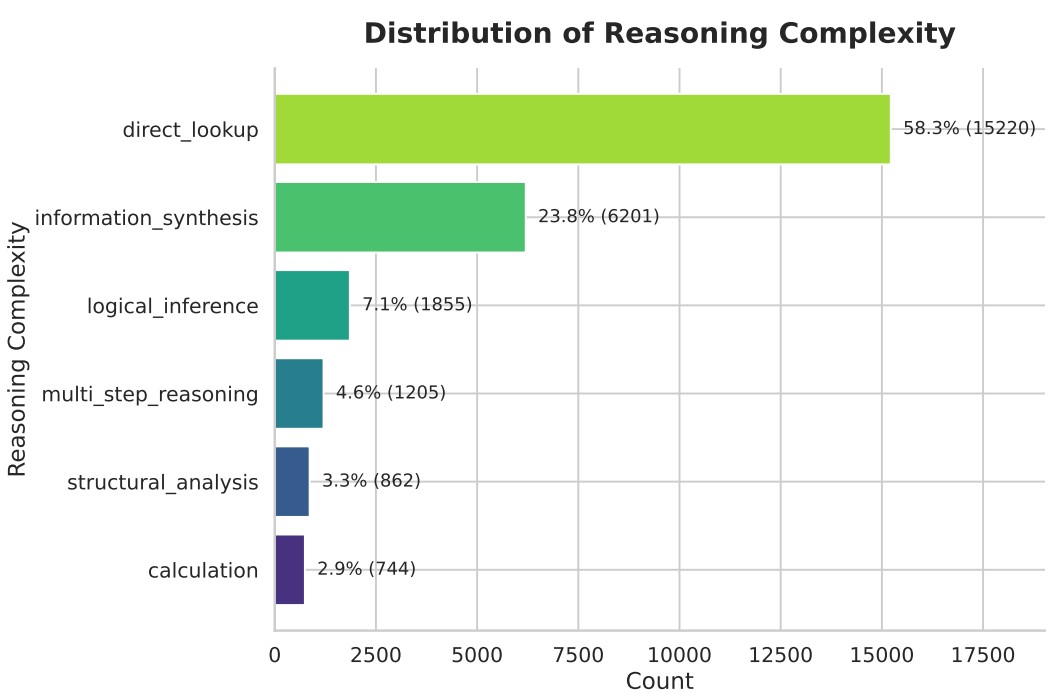

Figure 8: Distribution of reasoning complexity. A key feature of the dataset is that over 40% of questions require more than simple direct lookups, demanding skills like information synthesis (23.8%) and multi-step reasoning (4.6%) to arrive at the correct answer.

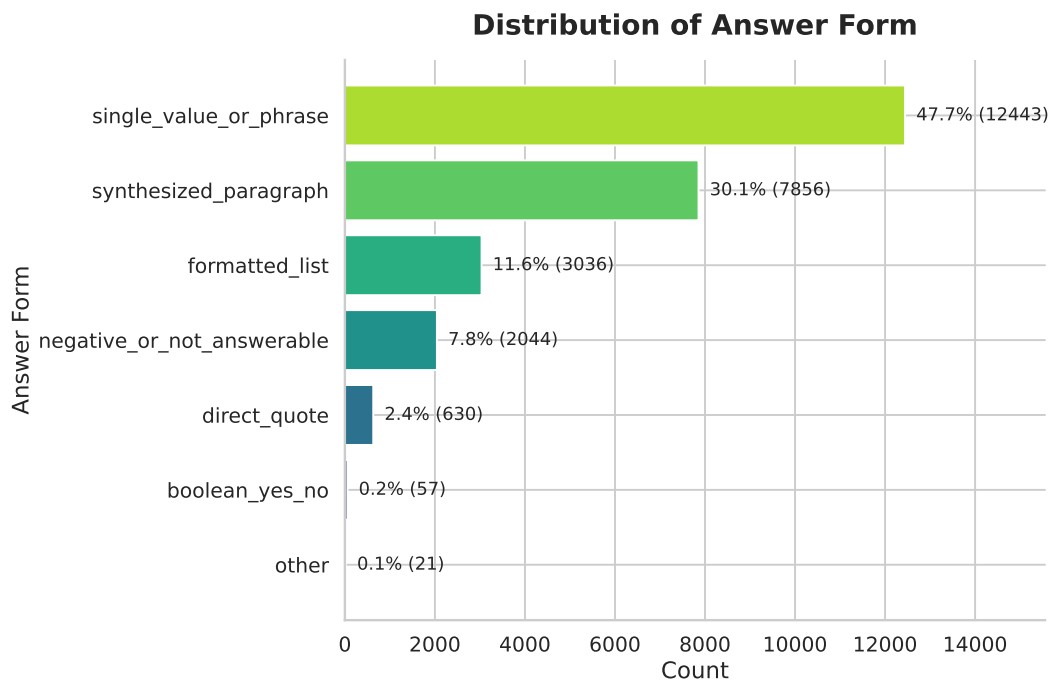

Figure 9: Distribution of expected answer forms. The dataset requires models to generate a variety of output formats, from concise single phrases (47.7%) to comprehensive synthesized paragraphs (30.1%), mirroring real-world application needs.

Table 6: Datasets used for Stage 1 foundational fine-tuning. The total volume comprises over 48 000 question-answer pairs, providing a solid foundation for the model.

| Dataset Component | Primary Source | Task Type | Size (Pairs) | Key Characteristics |
|---|---|---|---|---|
| ChartQA (subset) | Open-source | ChartQA | 5000 | Short-form question-answering pairs focused on chart comprehension. |
| DocVQA (subset) | Public Benchmark | DocVQA | 5349 | Question-answering on real-world scanned documents with challenging OCR. |
| Paper+CC VQA Mix | Scholarly Papers, CC | Mixed VQA | 2127 | A composite dataset blending academic paper content with web data from Common Crawl. |
| Curated DocQA Mix | Diverse Sources | Single-page QA | 29 489 | A large, diverse collection of QA pairs from various single-page document types. |
| Visual QA (generic) | Public VQA Dataset | General VQA | 6000 | Standard open-domain visual question-answering pairs to bolster general visual reasoning. |

Table 7: Summary statistics of the CoR-Dataset. The dataset is intentionally skewed towards more complex, multi-faceted categories to foster advanced reasoning capabilities.

| Dimension | Dominant Category | Count | Percentage |
|---|---|---|---|
| **Document Type** | Academic Paper | 14 417 | 55.3% |
| | *(Top 3 total)* | 21 634 | 82.9% |
| **Question Intent** | Factual Extraction | 18 562 | 71.2% |
| | *(Top 3 total)* | 22 465 | 86.1% |
| **Reasoning Complexity** | Direct Lookup | 15 220 | 58.3% |
| | Information Synthesis | 6201 | 23.8% |
| **Answer Form** | Single Value/Phrase | 12 443 | 47.7% |
| | Synthesized Paragraph | 7856 | 30.1% |

pushing models beyond simple lookups. Similarly, while direct lookups are common, over 40% of the questions require more advanced cognitive skills like information synthesis and multi-step reasoning. This composition ensures that models trained on the CoR-Dataset are robustly prepared for diverse and demanding document understanding scenarios.

## A.9 DETAILED CHAIN-OF-READING APPLICATION EXAMPLES

This section provides a series of detailed, step-by-step examples to illustrate the practical application and effectiveness of the Chain-of-Reading (CoR) framework. Each case breaks down a complex document understanding task into the CoR stages, demonstrating how the model navigates challenges like irregular layouts, repetitive content, and nested data structures. The document examples used in this paper are sourced from MMlongbench-doc(Ma et al., 2024), LongDoc-URL(Deng et al., 2024), and publicly available documents online intended for academic exchange.

### A.9.1 Example 1: Reasoning over Irregular Visual Layouts

This first example demonstrates CoR's capability to parse non-standard document layouts where data is presented in a visually associative manner rather than a conventional grid table. This requires the model to perform visuo-spatial reasoning to correctly link headers to data before performing any calculations.

**Question:** *"What is the absolute difference in the Solar Heat Gain Coefficient between the 'STANDARD LOW E²' and the 'ZO-E-SHIELD 5 EXTREME' window?"*

**Generated CoR Trace:**

> **[Task Planning]** The user wants to calculate the difference in the Solar Heat Gain Coefficient (SHGC) for two products. I will first locate the chart, then visually associate the product names with their data columns, extract the two SHGC values, and finally compute the difference.
>
> **[Phased & Focused Search]** I have identified the performance chart. "STANDARD LOW E²" is the first column; "ZO-E-SHIELD 5 EXTREME" is the third column. Now locating the "Solar Heat Gain Coefficient" row.
>
> **[Cross-modal Evidence Integration]** The value for 'Standard' (Column 1) is 0.30. The value for 'Extreme' (Column 3) is 0.19.
>
> **[Grounded Reasoning & Verification]** The task is to calculate the absolute difference: 0.30 - 0.19 = 0.11. The steps are verified, and the answer is grounded in the extracted values.

**Final Answer:** The absolute difference is **0.11**.

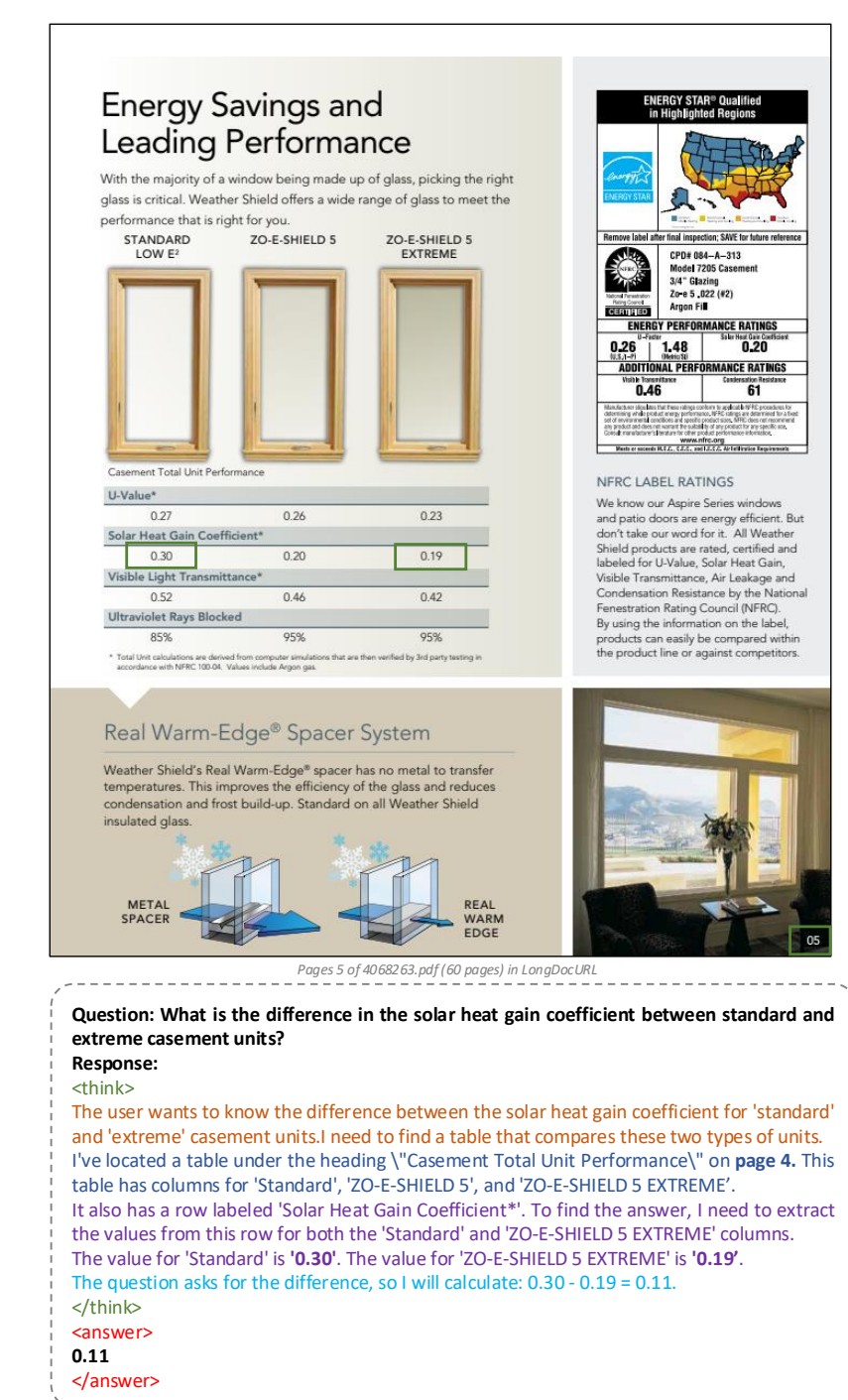

*Pages 5 of 4068263.pdf (60 pages) in LongDocURL*

**Question: What is the difference in the solar heat gain coefficient between standard and extreme casement units?**

**Response:**

<think>

The user wants to know the difference between the solar heat gain coefficient for 'standard' and 'extreme' casement units.I need to find a table that compares these two types of units. I've located a table under the heading \"Casement Total Unit Performance\" on **page 4.** This table has columns for 'Standard', 'ZO-E-SHIELD 5', and 'ZO-E-SHIELD 5 EXTREME'.

It also has a row labeled 'Solar Heat Gain Coefficient*'. To find the answer, I need to extract the values from this row for both the 'Standard' and 'ZO-E-SHIELD 5 EXTREME' columns.

The value for 'Standard' is **'0.30'**. The value for 'ZO-E-SHIELD 5 EXTREME' is **'0.19'**.

The question asks for the difference, so I will calculate: 0.30 - 0.19 = 0.11.

</think>

<answer>

**0.11**

</answer>

Figure 10: A product brochure snippet with a non-standard layout. The model must visually associate the product titles at the top with their corresponding data columns below.

### A.9.2    EXAMPLE 2: EVIDENCE LOCALIZATION IN REPETITIVE DOCUMENTS

This case showcases CoR's ability to navigate visually repetitive multi-page documents, a common scenario in reports and surveys. The key challenge is to disambiguate between many similar-looking

pages to pinpoint the precise evidence required by the user's query, thereby avoiding hallucinations from incorrect context.

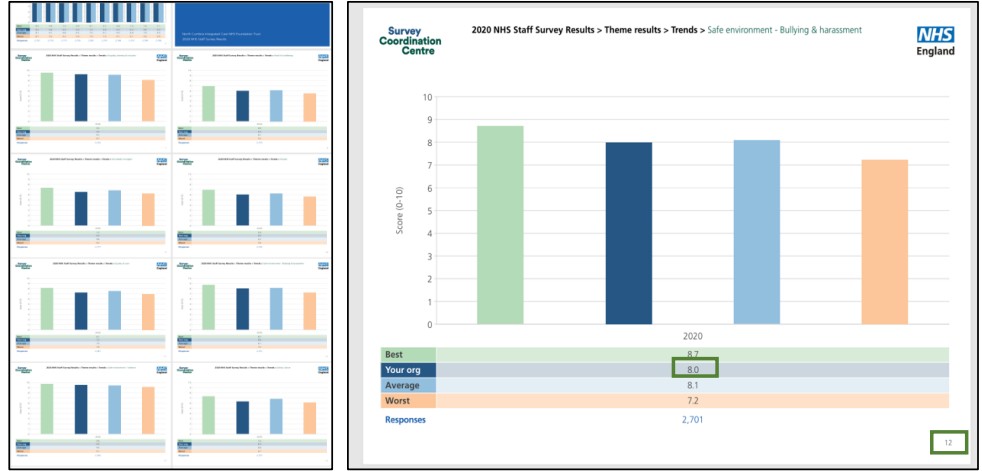

Pages 12 of 4083860.pdf (69 pages) in LongDocURL

**Question: What is the highest score for 'Safe environment - Bullying & harassment**
**Response:**
<think>
The user wants to know the "highest score" for the theme "Safe environment - Bullying & harassment". I need to find the table with this title.
I have located it on **page 12**.
The table has four rows: **'Best', 'Your org', 'Average'**, and **'Worst'** . The question asks for the highest score, which corresponds to the 'Best' row. I will extract the value from the 'Best' row.
The value is **'8.7'.**
</think>
<answer>
**8.7**
</answer>"

Figure 11: A multi-page report where many pages share an identical visual template. The main challenge is to locate the specific page that matches the query's context before data extraction.

**Question:** *"What is the highest score for 'Safe environment - Bullying & harassment'?"*

**Generated CoR Trace:**

> **[Task Planning]** The user wants the "highest score" for a specific theme. I need to scan the document for the theme title "Safe environment - Bullying & harassment" to find the correct page.
>
> **[Phased & Focused Search]** I am scanning the document thumbnails and titles. I have located the correct chart on **page 12**.
>
> **[Cross-modal Evidence Integration]** The question asks for the "highest score." In the table on page 12, this corresponds to the row labeled "Best". The value in this row is 8.7.
>
> **[Grounded Reasoning & Verification]** The evidence is located on the correct page and the semantic meaning of "highest score" is correctly mapped to "Best". The extracted value is verified.

**Final Answer:** The highest score is **8.7**.

### A.9.3 EXAMPLE 3: NAVIGATING COMPLEX NESTED TABLES

This final example highlights CoR's proficiency in parsing complex, nested tables commonly found in academic papers. The challenge lies not just in finding the correct table, but in navigating its hierarchical structure (Panels, Columns, Sub-columns, and Rows) to pinpoint a single data point with high precision.

**Question:** *"What is the regression discontinuity (RD) effect of stimulus payments on consumer spending in bottom income quartile ZIP codes over the period April 1-April 30 (Table 4, Panel A, Column 1)?"*

**Generated CoR Trace:**

> **[Task Planning]** The user is asking for a very specific data point. I will deconstruct the query into a navigation path: Table 4 -¿ Panel A -¿ Column "Bottom Income Quartile ZIP Codes" / Sub-column "(1)" -¿ Row "RD Effect of Stimulus:".
>
> **[Phased & Focused Search]** Navigating to Table 4, titled "Causal Effect of Stimulus Payments...". Focusing on "Panel A: Impact... on Consumer spending".
>
> **[Cross-modal Evidence Integration]** I have pinpointed the intersection of the specified row and column. The value at this exact location in the table is "25.15".
>
> **[Grounded Reasoning & Verification]** The extracted value "25.15" directly corresponds to the multi-coordinate location specified in the query. The retrieval path is verified.

**Final Answer:** The value is **25.15**.

**Table 4**

Causal Effect of Stimulus Payments on Spending and Small Business Revenue:
Regression Discontinuity Estimates

*Panel A: Impact of Stimulus Payments on Consumer Spending*

| Dep. Var.: | Change in Consumer Spending (%) | | | |
|---|---|---|---|---|
| | Bottom Income Quartile ZIP Codes | | Top Income Quartile ZIP Codes | |
| | (1) | (2) | (3) | (4) |
| RD Effect of Stimulus: | 25.15 | 36.97 | 8.45 | 15.83 |
| | (7.15) | (9.81) | (3.83) | (5.14) |
| Window: | April 1 - April 30 | April 7 - April 21 | April 1 - April 30 | April 7 - April 21 |

*Panel B: Impact of Stimulus Payments on Small Business Revenue*

| Dep. Var.: | Change in Small Business Revenue (%) | | | |
|---|---|---|---|---|
| | Bottom Rent Quartile ZIP Codes | | Top Rent Quartile ZIP Codes | |
| | (1) | (2) | (3) | (4) |
| RD Effect of Stimulus: | 17.92 | 20.83 | 1.20 | -7.54 |
| | (9.59) | (16.76) | (6.27) | (10.45) |
| Window: | April 1 - April 30 | April 7 - April 21 | April 1 - April 30 | April 7 - April 21 |

*Notes*: This table shows regression discontinuity estimates of changes in spending and business revenue around the date of stimulus payments on April 15, 2020. Panel A shows estimated effects of stimulus payments on consumer spending. To construct the estimates, we first express consumer spending on each day as a percentage change relative to mean daily consumer spending over the period January 4-31 in the corresponding calendar year. We then residualize these daily percentage changes with respect to day of week and first day of the month fixed effects, which we estimate using data from January 1, 2019, to May 10, 2019. We then compute OLS regressions of the residualized outcome variable on an indicator variable for the date being on or after April 15 2020, using a linear control function before and after April 15, and excluding the partially treated date of April 14. The first row shows the coefficient on the indicator variable for the date being on or after April 15, which we refer to as the RD effect of stimulus; standard errors are reported in parentheses. The dependent variable is scaled as a percentage change from January so that, for example, the first row of Column (1) indicates that stimulus payments increased consumer spending by 25.15 percentage points in bottom income quartile ZIP codes relative to the January 2020 level of spending. In columns (1) and (2), we compute daily changes in spending restricted to cardholders in ZIP codes in the bottom quartile of the distribution of ZIP code median household income (based on data from the 2014-2018 ACS). Columns (3) and (4) replicate columns (1) and (2), computing daily changes in spending restricted to cardholders living in the top income quartile of ZIP codes. The coefficient and standard error in columns (1) and (3) of Panel A correspond to the specifications displayed in Figures 13b and 13c. Panel B shows regression discontinuity estimates for the effect of stimulus payments on small business revenue using data from Womply. We first express small business revenue on each day relative to mean daily small business revenue over the period January 4-31 of the corresponding year. We then residualize daily changes in small business revenue as in Panel A, and compute OLS regressions as in Panel A. Columns (1) and (2) restrict to ZIP codes in the bottom quartile of the distribution of ZIP code median rent for a two bedroom apartment (based on data from the 2014-2018 ACS). Columns (3) and (4) restrict to businesses in the top rent quartile ZIP codes. The coefficient and standard error in columns (1) and (3) of Panel B correspond to the specifications displayed in Figures 14a and 14b. In both panels, columns (1) and (3) include all of April 2020 in the regression specification, while columns (2) and (4) restrict to a narrower bandwidth, within one week on either side of the stimulus payment date. Data sources: Panel A: Affinity Solutions; Panel B: Womply.

*Pages 70 of 4078345.pdf (109 pages) in LongDocURL*

---

**Question: What is the regression discontinuity (RD) effect of stimulus payments on consumer spending in bottom income quartile ZIP codes over the period April 1-April 30 (Table 4, Panel A, Column 1)?**

**Response:**

<think>

This is a precise data extraction question. I need to find the value at the intersection of a specific row and column in Table 4. First, I'll deconstruct the question. The key components are: "regression discontinuity (RD) effect," "stimulus payments," "consumer spending," "bottom income quartile ZIP codes," and "**Table 4, Panel A, Column 1.**" Next, I'll narrate my search. I will navigate to Table 4, which is titled "Causal Effect of Stimulus Payments on Spending and Small Business Revenue: Regression Discontinuity Estimates." The question specifies "Panel A: Impact of Stimulus Payments on Consumer Spending," so I will focus on that panel . Now, I'll demonstrate precision. Within Panel A, I need to find the column corresponding to "Bottom Income Quartile ZIP Codes" and the sub-column "(1)". Then, I must locate the row labeled "RD Effect of Stimulus:". The value at the exact intersection of this row and column is "25.15".

</think>

<answer>

**25.15**

</answer>

---

Figure 12: An academic table with a complex, hierarchical structure (Panels and multi-level columns). Precise extraction requires careful navigation of this structure.

### A.9.4 EXAMPLE 4: MULTI-STEP CALCULATION WITH PRIOR KNOWLEDGE

This example demonstrates the model's ability to solve complex questions that require not only precise data extraction but also the application of prior knowledge, such as a specific mathematical formula. The CoR process shows how the model formulates a calculation plan, extracts the necessary data points, and then executes the multi-step reasoning to arrive at the final answer.

**Question:** *"What was the percentage change in the Maltese lira against the euro between the average for Q4 2004 and the average for Q4 2003?"*

**Generated CoR Trace:**

> **[Task Planning]** This is a multi-step calculation question. My plan is: 1. Find the exchange rates for the Maltese lira against the euro for Q4 2004 and Q4 2003. 2. Apply the percentage change formula: ((New Value - Old Value) / Old Value) * 100.
>
> **[Phased & Focused Search]** I have located "Table 5.3 MALTESE LIRA EXCHANGE RATES...". I need to extract values from the 'EUR/MTL' column for the rows 'Average for Q4 2004' and 'Average for Q4 2003'.
>
> **[Cross-modal Evidence Integration]** The value for 'Average for Q4 2004' is 2.3181. The value for 'Average for Q4 2003' is 2.3327.
>
> **[Grounded Reasoning & Verification]** Now, I will perform the calculation based on my plan: ((2.3181 - 2.3327) / 2.3327) * 100. This equals (-0.0146 / 2.3327) * 100, which is approximately -0.626%. Rounding to one decimal place gives -0.6%. The calculation is verified.

**Final Answer:** The percentage change was **-0.6%**.

2003, net direct investment inflows dropped by Lm52.4 million to Lm39.9 million. This essentially reflected a smaller amount of inter-company transactions, which include trade credits and loans between parents and subsidiaries. Moreover, net inflows of equity capital, which had been boosted by heavy investment in one bank in the fourth quarter of 2003, also declined. These developments outweighed a rise in earnings retained by foreign-owned firms operating in Malta. Net flows related to financial derivatives, which record instruments including options, futures and swaps, also turned negative.

Conversely, net portfolio outflows dropped sharply, from Lm104.5 million to Lm17.9 million. Although the banking sector continued to invest

**The Maltese lira**

During the fourth quarter of 2004, the Maltese lira continued to depreciate against the euro, shedding 1.3% (see Table 5.3). On the other hand, the lira maintained its appreciation against the US dollar, gaining 8.5% as a result of the continuous retreat of the dollar on international foreign exchange markets. The lira also continued to strengthen against sterling and the Japanese yen, appreciating by 1.5% and 0.7%, respectively. Throughout the first quarter of 2005, counter to the previous trend, the lira rose marginally against the euro and depreciated against the dollar (see Chart 5.2). At the same time, the lira rose further against the yen, while it fell against sterling.

**Table 5.3**

**MALTESE LIRA EXCHANGE RATES AGAINST SELECTED CURRENCIES**

| Period | EUR/MTL | USD/MTL | GBP/MTL | JPY/MTL |
| --- | --- | --- | --- | --- |
| Average for Q4 2004 | 2.3181 | 3.0053 | 1.6114 | 317.8 |
| Average for Q4 2003 | 2.3327 | 2.7747 | 1.6269 | 302.0 |
| % change | -0.6 | 8.3 | -1.0 | 5.2 |
| Closing rate on 31.12.2004 | 2.3023 | 3.1393 | 1.6252 | 321.7 |
| Closing rate on 30.09.2004 | 2.3315 | 2.8921 | 1.6012 | 319.6 |
| % change | -1.3 | 8.5 | 1.5 | 0.7 |

*Pages 46 of 4064830.pdf (133 pages) in LongDocURL*

**Question: What was the percentage change in the Maltese lira against the euro between the average for Q4 2004 and the average for Q4 2003?**
**Response:**
<think>
This is a multi-step calculation question. First, I need to deconstruct it. I need to find the exchange rate for the Maltese lira against the euro for two periods: **Q4 2004** and **Q4 2003.** Second, I need to calculate the percentage change using the formula: ((New Value - Old Value) / Old Value) * 100. I'll search for a table containing exchange rates. I've located \"Table 5.3 MALTESE LIRA EXCHANGE RATES AGAINST SELEcTcD CURRENCIES\". I need to be precise and extract the values from the 'EUR/MTL' column. For the 'Average for Q4 2004' row, the value is **2.3181**. For the 'Average for Q4 2003' row, the value is **2.3327**. Now I'll perform the calculation: ((2.3181 - 2.3327) / 2.3327) * 100. This simplifies to (-0.0146 / 2.3327) * 100, which equals -0.6%. The answer should be rounded to one decimal place.
</think>
<answer>
**-0.6%**
</answer>

Figure 13: A financial data table listing currency exchange rates. Answering questions based on this table requires finding multiple values and performing calculations.

### A.9.5 Example 5: Cross-Page Evidence Synthesis from Heterogeneous Layouts

This case demonstrates a crucial capability for real-world document analysis: synthesizing information scattered across disparate, non-consecutive pages with heterogeneous layouts, a common feature of slide decks and reports. The model must independently locate two pieces of evidence from visually distinct contexts and then perform a comparison.

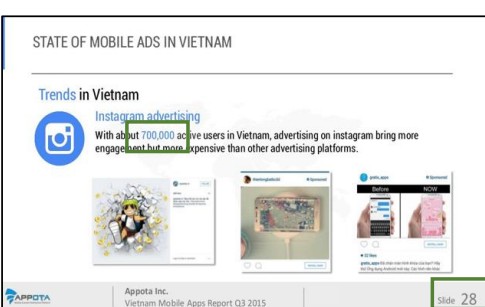 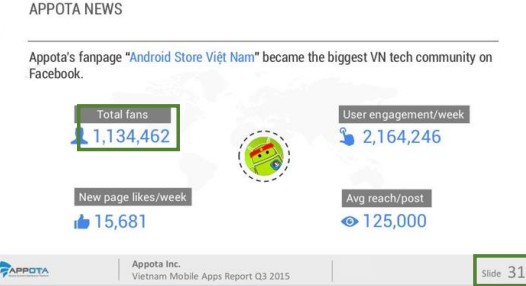

*Pages 25, 31 of reportq32015-151009093138-lva1-app6891_95.pdf (34pages) in MMLongBench-Doc*

**Question: As of Q3 2015, are there more active Instagram users in Vietnam, or are there more fans for Appota's fan page? Directly answer 'Instagram' or 'Appota '.**
**Response:**
<think>
The user is asking a direct comparison question about the number of users for two entities: "Instagram" and "Appota's fan page"
The document is a report titled "VIETNAM MOBILE APPS REPORT Q3 2015". This title is present on Slide 1, which is the **first page** of the document. This confirms the context and time frame of the report.
The question asks for a comparison as of Q3 2015. The document's title explicitly states this timeframe. I will search the document for the term "Instagram". **On Slide 28**, under the section "Trends in Vietnam", there is a sub-section "Instagram advertising". The text within this sub-section provides a specific figure: "With about 700,000 active users in Vietnam...". This directly answers the part of the question concerning Instagram users.Next, I will search the document for information related to "Appota's fan page". **On Slide 31**, under the section "APPOTA NEWS", there is a sub-section that discusses the fan page. The text reads: "Appota's fanpage "Android Store Việt Nam" became the biggest VN tech community on Facebook." This identifies the fan page. The slide also contains a table with the heading "Total fans" and the value "1,134,462". This is the number of total fans for the fan page.
Now, I will compare the two figures found in the document:
*Instagram users: 700,000 (Slide 28)
*Appota's fan page total fans: 1,134,462 (Slide 31)
Comparing these two numbers, 1,134,462 is greater than 700,000. Therefore, the number of fans for Appota's fan page is greater than the number of active Instagram users in Vietnam as of Q3 2015.n9. The question asks me to directly answer 'Instagram' or 'Appota'. Based on my analysis, the correct response is 'Appota'.
</think>
<answer>
**Appota**
</answer>

Figure 14: Evidence scattered across two different pages (Slide 28 and Slide 31) from a presentation. The pages have completely different visual styles, requiring the model to perform robust, context-independent searches.

**Question:** *"As of Q3 2015, are there more active Instagram users in Vietnam, or are there more fans for Appota's fan page? Directly answer 'Instagram' or 'Appota'."*

**Generated CoR Trace:**

> **[Task Planning]** This is a comparison question requiring two data points from different parts of the document. My plan is: 1. Search for the number of "Instagram users in Vietnam". 2. Search for the number of fans for "Appota's fan page". 3. Compare the two numbers and provide a direct answer as requested.
>
> **[Phased & Focused Search]** I am searching the document for "Instagram". On Slide 28, I found the text "With about 700,000 active users in Vietnam...". Next, I am searching for "Appota's fan page". On Slide 31, I found a table with "Total fans" and the value "1,134,462".
>
> **[Cross-modal Evidence Integration]** - Evidence 1 (Slide 28): Instagram active users = 700,000. - Evidence 2 (Slide 31): Appota's fan page fans = 1,134,462.
>
> **[Grounded Reasoning & Verification]** Now I will compare the two values. 1,134,462 is greater than 700,000. Therefore, Appota has more fans. The question requires a direct answer of 'Instagram' or 'Appota'. Based on my reasoning, the correct answer is 'Appota'.

**Final Answer: Appota**

A.9.6 EXAMPLE 6: LOGICAL REASONING ON VISUALLY ENCODED INFORMATION

This final case study demonstrates a sophisticated combination of skills: understanding implicit information from visual layouts, performing robust multilingual OCR, and executing logical set operations. The model must infer rankings from a visual list, correctly read non-English text (in this case, Vietnamese), and then perform a "set difference" comparison across two pages to find the unique item.

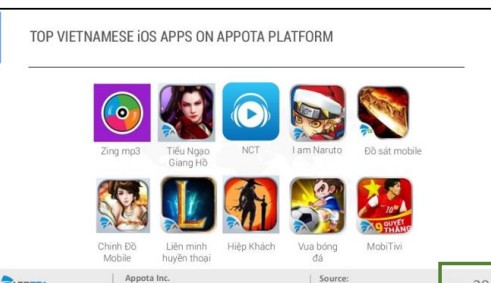 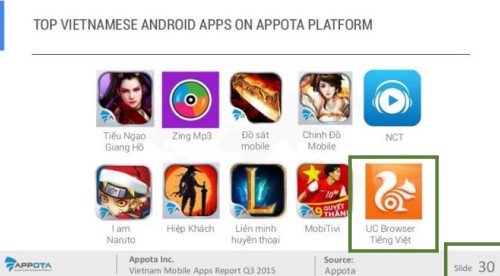

*Pages 29, 30 of reportq32015-151009093138-lva1-app6891_95.pdf (34pages) in MMLongBench-Doc*

**Question: Which APP on APPOTA platform is top 10 Vietnam Android App, but not top 10 Vietnam iOS App?**
**Response:**
<think>
The user is asking for an app that is in the top 10 on the "APPOTA platform" for "Vietnam Android App" but not for "Vietnam iOS App", which requires performing a logical set difference operation. My process begins by locating the two required charts within the document.

I have found the "TOP VIETNAMESE iOS APPS ON APPOTA PLATFORM" chart on **page 29** and the "TOP VIETNAMESE ANDROID APPS ON APPOTA PLATFORM" chart on **page 30**. A key challenge is that the rankings are not numbered but are implicitly conveyed by the top-to-bottom visual order, which I must correctly interpret.

After performing multilingual OCR, I have extracted the full iOS Top 10 list as **[Zing mp3, Tiểu Ngạo Giang Hồ, NCT, I am Naruto, Đồ sát mobile, Chinh Đồ Mobile, Liên minh huyền thoại, Hiệp Khách, Vua bóng đá, MobiTivi]** and the full Android Top 10 list as **[Tiểu Ngạo Giang Hồ, Zing Mp3, Đồ sát mobile, Chinh Đồ Mobile, NCT, I am Naruto, Liên minh huyền thoại, Hiệp Khách, MobiTivi, UC Browser Tiếng Việt]**. Now, I will execute the planned set difference by systematically comparing the Android list against the iOS list. My verification confirms that nine of the applications appear in both lists. The only app that is present in the Android list but is absent from the iOS list is "UC Browser Tiếng Việt"

Therefore, this is the unique and correct answer that satisfies all conditions of the user's query.
</think>
<answer>
**UC Browser Tiếng Việt**
</answer>

Figure 15: Two top-app charts from consecutive slides (29 and 30). The rankings are implicitly encoded by vertical position, and the app names are in Vietnamese, requiring advanced visual parsing and multilingual OCR capabilities.

**Question:** *"Which APP on APPOTA platform is top 10 Vietnam Android App, but not top 10 Vietnam iOS App?"*

**Generated CoR Trace:**

**[Task Planning]** The user is asking for an app that meets two criteria: it is in the Android top 10, but not in the iOS top 10. This requires a set difference operation. My plan is to: 1. Locate and extract the list of top 10 Android apps. 2. Locate and extract the list of top 10 iOS apps. 3. Compare the two lists to find the app present only in the Android list.

**[Phased & Focused Search]** I have located the "TOP VIETNAMESE iOS APPS ON APPOTA PLATFORM" chart on page 29. On page 30, I found the "TOP VIETNAMESE ANDROID APPS ON APPOTA PLATFORM" chart. The rankings are presented visually, from top to bottom.

**[Cross-modal Evidence Integration]** The model extracts the two lists of app names from the charts. The first list contains the top 10 iOS apps, and the second list contains the top 10 Android apps, including one app not present in the first list.

**[Grounded Reasoning & Verification]** Now I will perform the set difference operation: (Android List) - (iOS List). By systematically comparing the two extracted lists, I have determined that a specific browser app is present in the Android list but is absent from the iOS list. All other apps in the Android top 10 are also present in the iOS list. Therefore, this is the unique app that satisfies the user's condition.

**Final Answer: UC Browser Tieng Viet**

A.9.7    EXAMPLE 7: JOINT REASONING OVER TEXT AND UNCAPTIONED IMAGES

This example demonstrates the model's ability to perform joint reasoning by synthesizing information from both surrounding text and an uncaptioned, illustrative image. To answer the question, the model must first locate a relevant text passage and then use the context provided by that text to correctly interpret the visual information in the adjacent figure.

**Question:** *"What is the color on the watch that shows the aerobic zone?"*

**Generated CoR Trace:**

**[Task Planning]** The user is asking for the color corresponding to the "aerobic zone". My plan is to search the document for the term "aerobic zone" and analyze the surrounding text and any associated images for color information.

**[Phased & Focused Search]** I have searched the document and located the term "aerobic" on page 19, within section "3.1 Monitoring your heart rate". The text nearby mentions "Your watch will display different colors when your heart rate reaches corresponding zones..." and points to an example figure on the same page showing an outdoor run.

**[Cross-modal Evidence Integration]** - **Textual Evidence (Page 19):** Confirms that different zones are represented by different colors and directs attention to the figure. - **Visual Evidence (Page 19):** The figure, though uncaptioned, shows a watch interface. I can visually identify a section highlighted in yellow with the text label "Aerobic" directly next to it.

**[Grounded Reasoning & Verification]** By combining the textual clue (zones have colors) with the direct visual evidence (the label "Aerobic" is next to the color yellow), I can confidently conclude that the color for the aerobic zone is yellow. The reasoning is grounded in this direct text-to-image link.

**Final Answer: Yellow**

A.9.8    EXAMPLE 8: LINGUISTIC AND VISUAL REASONING ON SCANNED ARCHIVES

This example showcases the model's robustness and deep reasoning capabilities when dealing with low-quality, historical documents. The task requires accurate localization within the document, advanced OCR on degraded, archaic fonts, and a nuanced linguistic understanding to differentiate between singular and plural nouns in the figure captions to arrive at a correct count.

**Setting the heart rate zone calculation method**

The heart rate interval can be calculated based on the maximum heart rate percentage or HRR percentage. To set the heart rate interval calculation method, open the Huawei Health app, go to **Me > Settings > Heart rate limit and zones** and set **Calculation method** to either **Maximum heart rate percentage** or **HRR percentage**.

📖 NOTE

- If you select **Maximum heart rate percentage** as the calculation method, the heart rate zone for different types of workout activities (Extreme, Anaerobic, Aerobic, Fat-burning, and Warm-up) is calculated based on your maximum heart rate ("220 – your age" by default). Heart rate = Maximum heart rate x Maximum heart rate percentage.
- If you select HRR percentage as the calculation method, the heart rate interval for different types of workout activities (Advanced anaerobic, Basic anaerobic, Lactic acid, Advanced aerobic, and Basic aerobic) is calculated based on your heart rate reserve (HRmax - HRrest). Heart rate = Heart rate reserve x Heart rate reserve percentage + Resting heart rate.
- The heart rate zone calculation methods while you are running are not affected by the settings in the Huawei Health app. For most running courses, **HRR percentage** is selected by default.

Your watch will display different colors when your heart rate reaches corresponding zones during a workout. The following figure shows how heart rate is displayed during an outdoor run.

Aerobic
146
Heart rate(bpm)
6'30"  |  4.03
Pace(/km)   Distance(km)
00:25:30
Time
07:16

**Measuring your heart rate during a workout**

1. After you start a workout, swipe on the watch screen to check your real-time heart rate and heart rate zone.
2. After completing your workout, you can check your average heart rate, maximum heart rate, and heart rate zone on the workout results screen.

*Pages 23 of honor_watch_gs_pro.pdf (42pages) in MMLongBench-Doc*

**Question: What is the color on the watch that shows the aerobic zone?**
**Response:**
<think>
The user is asking for the specific color of the "aerobic zone" on the watch. I will search the entire document for the term "aerobic zone".
On **page 19**, under the section "3.1 Monitoring your heart rate", there is a sub-section titled "Setting the heart rate zone calculation method". Within this section, the text states: "Your watch will display different colors when your heart rate reaches corresponding zones during a workout."
It then provides an example image: "The following figure shows how heart rate is displayed during an outdoor run." This image is on the same page (19). The image explicitly labels the "Aerobic" zone with a yellow color. The document provides a direct visual example of the color associated with the aerobic zone. Therefore, the information is present and can be extracted directly from the image.The question asks for the color, which is a factual detail. "
The document provides this detail through its figures and text. The user's question is answerable.n9. The answer is "Yellow"
</think>
<answer>
**Yellow**
</answer>

Figure 16: An excerpt from a user manual on page 19. The answer is not in the text alone but requires linking the description of heart rate zones to the colors shown in the unlabeled example image.

**Question:** *"Among figure 1-4, how many figures show more than one breccia gash?"*

**Generated CoR Trace:**

> **[Task Planning]** The user wants me to count how many of the first four figures depict more than one "breccia gash". My plan is to locate the specified figures, carefully read the caption for each one, and use linguistic cues (singular vs. plural) to determine the count for each figure before summing the result.
>
> **[Phased & Focused Search]** I have located the relevant content on page 3 of the document. This page contains four diagrams labeled Fig. 1, Fig. 2, Fig. 3, and Fig. 4, each with a descriptive caption below it.
>
> **[Cross-modal Evidence Integration]** I will analyze each figure's caption: - **Fig. 1:** The caption reads "Diagram of Breccia-Gash..." (singular). This figure shows one gash. - **Fig. 2:** The caption reads "Diagram of Breccia Gashes..." (plural). This figure shows more than one gash. - **Fig. 3:** The caption reads "Diagram of narrow Breccia Gashes" (plural). This figure shows more than one gash. - **Fig. 4:** The caption reads "Diagram of Breccia-Gash..." (singular). This figure shows one gash.
>
> **[Grounded Reasoning & Verification]** Based on the linguistic analysis of the captions, Figures 2 and 3 show "more than one breccia gash". Figures 1 and 4 show only one. Therefore, the total count of figures showing more than one gash is two. The reasoning is grounded in the grammatical distinction found in the scanned text.

**Final Answer: 2**

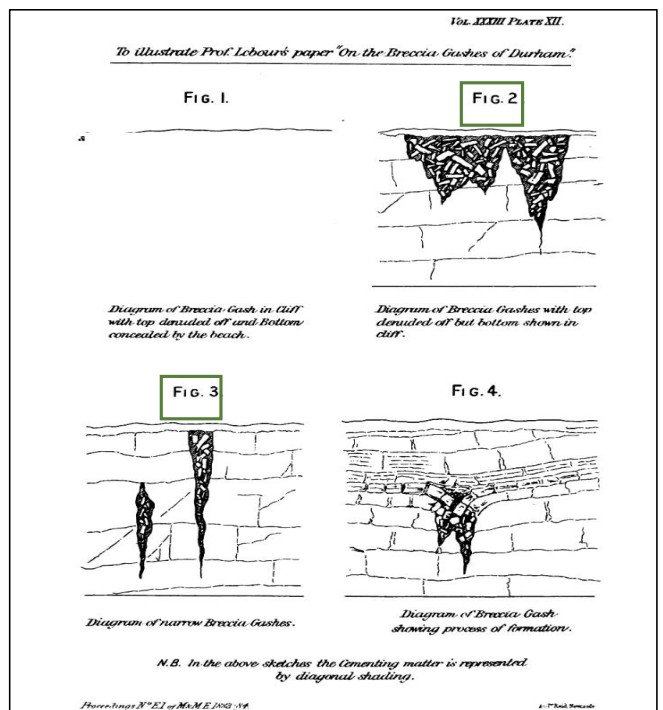

*Pages 3 of fd76bbefe469561966e5387aa709c482.pdf (17 pages) in MMLongBench-Doc*

**Question: Among figure 1-4, how many figures show more than one breccia gash?**
**Response:**

<think>

The user's question asks to count the figures that depict more than one breccia gash among figures 1-4. I will examine the provided document images to locate the figures.

**Page 3** contains the figures. The caption for this page reads: "Vol. XXXIII PLATE XII. To illustrate Prof. Lebour's paper 'On the Breccia Gashes of Durham'." The figures are numbered as follows: **\* Fig. 1: "Diagram of Breccia Gash in CLIP with top denuded off and Bottom concealed by the beach." \* Fig. 2: "Diagram of Breccia Gashes with top denuded off but bottom shown in cliff." \* Fig. 3: "Diagram of narrow Breccia Gashes." \* Fig. 4: "Diagram of Breccia Gash showing process of formation."**

The question specifies "figure 1-4". This means I will analyze figures 1, 2, 3, and 4. Now, I will examine each figure to determine if it shows more than one breccia gash. \* Figure 1: This figure shows a single diagram of a "Breccia Gash". \* Figure 2: This figure shows a diagram of "Breccia Gashes". The word "Gashes" implies multiple gashes are depicted. The caption explicitly says "...but bottom shown in cliff." This confirms it shows more than one gash. \* Figure 3: This figure shows diagrams of "narrow Breccia Gashes". The word "narrow" does not change the fact that they are multiple gashes. The caption explicitly says "...of narrow Breccia Gashes." This confirms it shows more than one gash. \* Figure 4: This figure shows a diagram of a "Breccia Gash".

It does not show more than one gash. Based on this analysis, figures 2 and 3 show more than one breccia gash. The question asks for the total count. Therefore, the answer is 2.

</think>

<answer>

**2**

</answer>

Figure 17: A low-quality scan from a historical scientific paper, identified by the model as page 3. To answer the question, the model must accurately read the captions for Figures 1-4 and interpret the linguistic cues within them.

### A.9.9 EXAMPLE 9: MULTI-STEP REASONING ACROSS HETEROGENEOUS PAGES

This case study demonstrates the model's ability to solve complex, multi-step questions by synthesizing clues from disparate sections of a document. The process requires decomposing the initial query, locating evidence across heterogeneous page types (a table and a chart), interpreting graphical data, and performing a final calculation.

**Question:** *"In the year in which Palestine was added to the survey, respondents who believe traditional values are outdated increased by how many percentage points compared to 2011?"*

**Generated CoR Trace:**

> **[Task Planning]** The user's question requires a multi-step process. First, I need to identify the year Palestine was added to the survey. Second, I must find the percentage point data for that year and for 2011. Finally, I will calculate the difference.
>
> **[Phased & Focused Search]** I am searching for "Palestine". On page 6, a table lists "Palestine" under the heading "New in 2014". This establishes the target year. Next, I am locating the data on values. On page 10, a bar chart provides the percentage of respondents who believe "Traditional values are outdated" for various years.
>
> **[Cross-modal Evidence Integration]** - **Evidence 1 (from Table, Page 6):** The year Palestine was added is 2014. - **Evidence 2 (from Chart, Page 10):** The value for 2011 is 17%. - **Evidence 3 (from Chart, Page 10):** The value for 2014 is 46%.
>
> **[Grounded Reasoning & Verification]** The question asks for the increase in percentage points. The calculation is a simple subtraction: 46% (the value for 2014) minus 17% (the value for 2011). The result is 29 percentage points. The reasoning is verified, as I correctly linked the country to the year and then used that year to extract the correct data for the final calculation.

**Final Answer: 29**

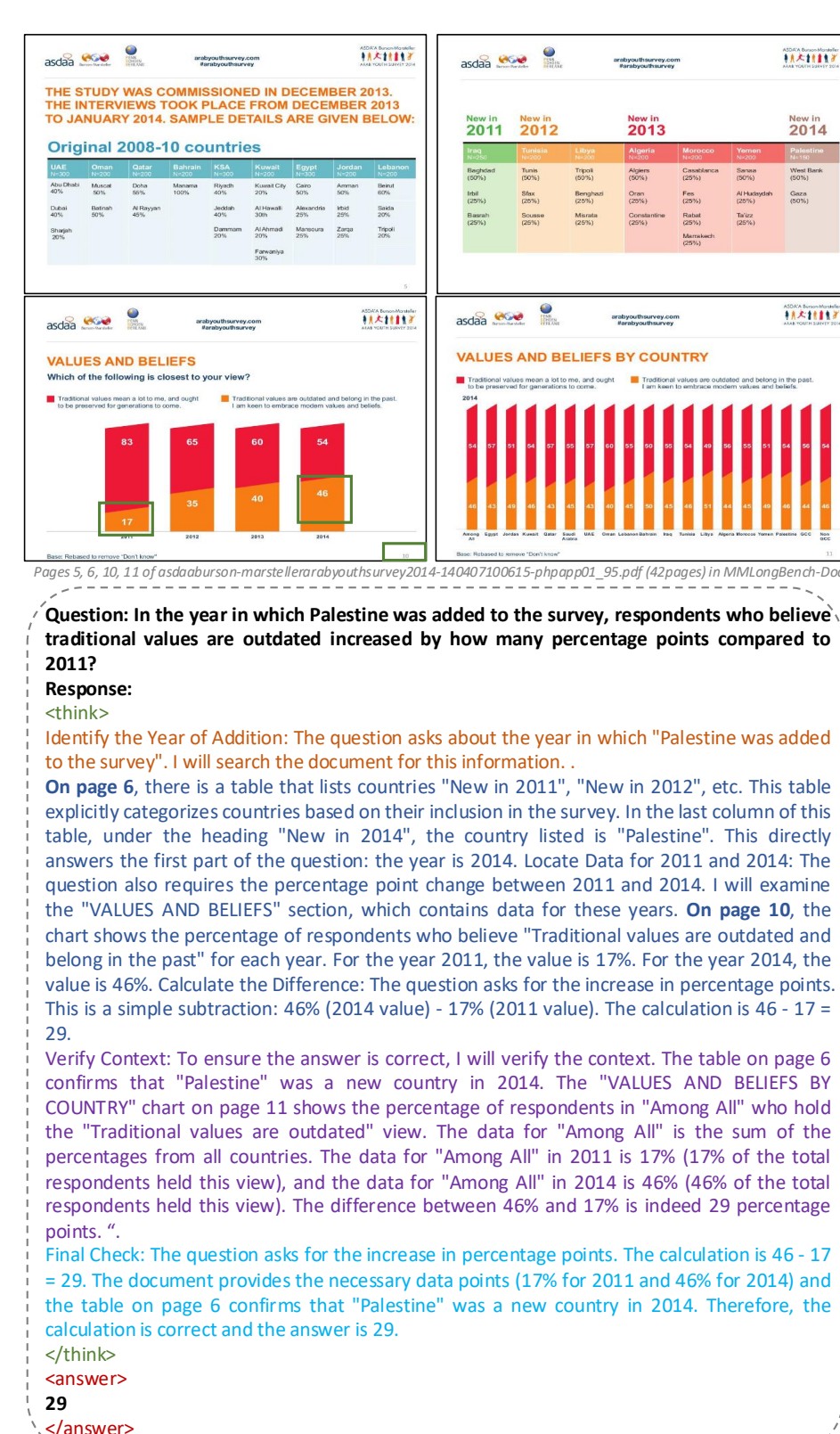

Pages 5, 6, 10, 11 of asdaaburson-marstellerarabyouthsurvey2014-140407100615-phpapp01_95.pdf (42pages) in MMLongBench-Doc

**Question:** In the year in which Palestine was added to the survey, respondents who believe traditional values are outdated increased by how many percentage points compared to 2011?

**Response:**

<think>

Identify the Year of Addition: The question asks about the year in which "Palestine was added to the survey". I will search the document for this information. .

**On page 6**, there is a table that lists countries "New in 2011", "New in 2012", etc. This table explicitly categorizes countries based on their inclusion in the survey. In the last column of this table, under the heading "New in 2014", the country listed is "Palestine". This directly answers the first part of the question: the year is 2014. Locate Data for 2011 and 2014: The question also requires the percentage point change between 2011 and 2014. I will examine the "VALUES AND BELIEFS" section, which contains data for these years. **On page 10**, the chart shows the percentage of respondents who believe "Traditional values are outdated and belong in the past" for each year. For the year 2011, the value is 17%. For the year 2014, the value is 46%. Calculate the Difference: The question asks for the increase in percentage points. This is a simple subtraction: 46% (2014 value) - 17% (2011 value). The calculation is 46 - 17 = 29.

Verify Context: To ensure the answer is correct, I will verify the context. The table on page 6 confirms that "Palestine" was a new country in 2014. The "VALUES AND BELIEFS BY COUNTRY" chart on page 11 shows the percentage of respondents in "Among All" who hold the "Traditional values are outdated" view. The data for "Among All" is the sum of the percentages from all countries. The data for "Among All" in 2011 is 17% (17% of the total respondents held this view), and the data for "Among All" in 2014 is 46% (46% of the total respondents held this view). The difference between 46% and 17% is indeed 29 percentage points. ".

Final Check: The question asks for the increase in percentage points. The calculation is 46 - 17 = 29. The document provides the necessary data points (17% for 2011 and 46% for 2014) and the table on page 6 confirms that "Palestine" was a new country in 2014. Therefore, the calculation is correct and the answer is 29.

</think>

<answer>

**29**

</answer>

Figure 18: Evidence required to solve the query is scattered across two different pages. The model must first use the table on page 6 to identify the target year (2014), and then use that information to extract relevant data from the bar chart on page 10.

A.9.10   EXAMPLE 10: ROBUSTNESS ON A HIGH-DIFFICULTY, ERROR-PRONE TASK

This case study demonstrates the model's robustness and high precision on a complex, multi-constraint query. The inherent difficulty of this task is underscored by the fact that the benchmark's own annotations for the evidence pages were found to be incorrect, likely due to the challenge of pinpointing specific data across multiple, similarly formatted tables. Despite this, our model successfully navigated the document and independently located the correct evidence on pages 15 and 16. This success on an error-prone task highlights the model's superior ability to deconstruct complex questions and maintain reasoning accuracy, a key advantage of the CoR framework.

**Question:** *"What is the percentage gap between the percentage of the male 65+ age group who use the internet or email, and the percentage of that group who have broadband at home, according to the Pew Research Center's Internet Project July 18-September 30, 2013 tracking survey?"*

**Generated CoR Trace:**

> **[Task Planning]** The user is asking for a "percentage gap", which implies subtraction. I need to find two specific percentages for the "male 65+" group from a specific survey: their internet usage and their broadband adoption rate. My plan is to locate both values and then calculate the difference.
>
> **[Phased & Focused Search]** I am searching for tables related to the "Pew Research Center... 2013 tracking survey". I have located a table for "Internet usage for older adults" on page 15, and another for "Broadband adoption for older adults" on page 16. In both tables, I will find the row for "Male".
>
> **[Cross-modal Evidence Integration]** - **Evidence 1 (from Table, Page 15):** In the "Internet usage" table, the value for "Male (n=612)" is 65(%). - **Evidence 2 (from Table, Page 16):** In the "Broadband adoption" table, the value for "Male (n=612)" is 53(%).
>
> **[Grounded Reasoning & Verification]** The task is to calculate the percentage gap. The calculation is 65% (internet usage) - 53% (broadband adoption). The result is 12 percentage points. The model's ability to independently locate the correct evidence pages (15 and 16) confirms its robust and accurate understanding of the document's content and structure.

**Final Answer: 12**

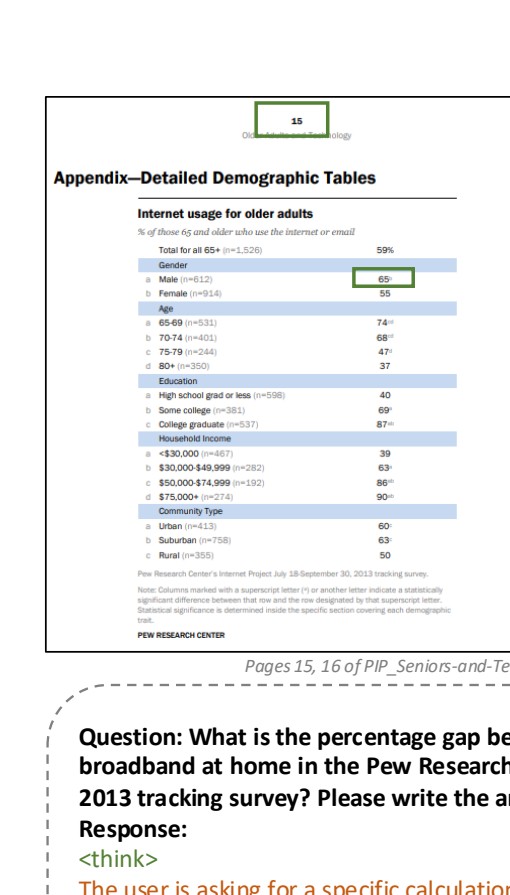

*Pages 15, 16 of PIP_Seniors-and-Tech-Use_040314.pdf (27pages) in MMLongBench-Doc*

**Question: What is the percentage gap between male 65+ age group who use internet and broadband at home in the Pew Research Center's Internet Project July 18-September 30, 2013 tracking survey? Please write the answer into float format.**
**Response:**
<think>
The user is asking for a specific calculation: the percentage gap between two groups ("male 65+ age group") and two metrics ("internet" and "broadband at home") from a specific survey ("Pew Research Center's Internet Project July 18-September 30, 2013 tracking survey"). I will search the document for the keywords "male", "65+", "internet", "broadband", and the specified survey date range.

**On page 15**, under the table titled "Internet usage for older adults", I find the row "Gender" and its sub-row "Male (n=612)". The value in this cell is "65a". This represents the percentage of male seniors who use the internet or email. **On page 16**, under the table titled "Broadband adoption for older adults", I find the row "Gender" and its sub-row "Male (n=612)". The value in this cell is "53b". This represents the percentage of male seniors who have broadband at home. The question asks for the "percentage gap". This means I need to calculate the difference between these two percentages. .

The calculation is: **65% (internet usage) - 53% (broadband at home) = 12%.** The question also specifies that the answer should be written in "float format". The number 12 is already a float. The final check is the context. The tables are explicitly for the "Pew Research Center's Internet Project July 18-September 30, 2013 tracking survey".

The text on page 15 and page 16 confirms this survey period. The calculated value directly answers the question.
</think>
<answer>
**12**
</answer>

Figure 19: Evidence for the query is split across two tables on consecutive pages (15 and 16). The model must correctly parse the complex query to locate the precise cells corresponding to "Male 65+" for both "internet usage" and "broadband adoption".

### A.9.11 EXAMPLE 11: IDENTIFYING AND REJECTING HALLUCINATION TRAPS

This final, crucial example demonstrates the model's advanced ability to perform meta-reasoning and resist "hallucination traps"—questions that are based on a false premise about the document's content. Instead of attempting to generate a factually incorrect answer, the model correctly identifies the discrepancy by performing a comprehensive document search, locating the most relevant figures, and concluding that the question's core subject ("BERT model") is not present. This capability is vital for building trustworthy and reliable document analysis systems.

**Question:** *"In the pipeline diagram of the BERT model, which type of organism is used as the input case?"*

**Generated CoR Trace:**

> **[Task Planning]** The user is asking about an input case in the "pipeline diagram of the BERT model". My plan is to first search the entire document for any mention of a "BERT model" pipeline diagram. If found, I will analyze it. If not found, I must verify if the document discusses BERT at all to determine if the question is answerable.
>
> **[Phased & Focused Search]** A global search for "BERT" yields no results. However, to be thorough, I searched for "pipeline diagram". This led me to two key figures: Fig. 1 on page 2 and Fig. 2 on page 6.
>
> **[Cross-modal Evidence Integration]** - **Fig. 1 (Page 2):** The caption discusses the motivation for "CLIP and MLLM" and the proposed "RAR" model. No mention of BERT. - **Fig. 2 (Page 6):** The caption explicitly describes the "Pipeline of RAR". It details a multimodal retriever and refiner. No mention of BERT.
>
> **[Grounded Reasoning & Verification]** My comprehensive search confirms that the document does not contain a pipeline diagram for the BERT model; the diagrams present are for different models (RAR, MLLM). The user's question is based on a false premise. Therefore, the question is unanswerable from the given document. Attempting to answer it would lead to hallucination.

**Final Answer: The provided document does not contain a pipeline diagram for the BERT model.**

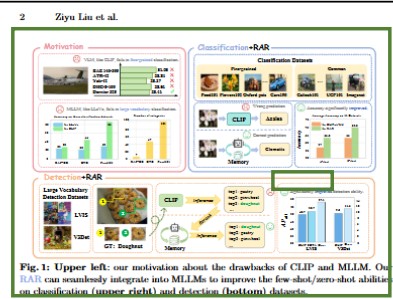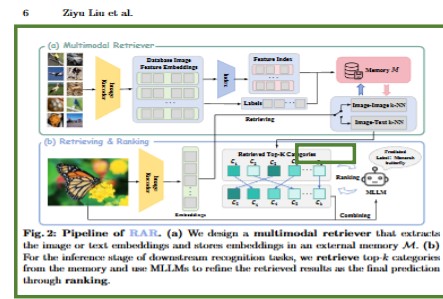

*Pages 2, 6 of RAR.pdf (28 pages) in MMLongBench-Doc*

---

**Question: In the pipeline diagram of the BERT model, which type of organism is used as the input case?**

**Response:**

<think>

The user is asking about the input organism in the pipeline diagram of the BERT model. The document is titled "RAR: RRetrieving And Ranking Augmented MLLMs for Visual Recognition". This title does not mention "BERT" or a "pipeline diagram". The user has provided two document images. I will examine these images to find the relevant information. Image 1 (page 2) contains a figure caption that reads: "Fig. 1: Upper left: our motivation about the drawbacks of CLIP and MLLM. Our RAR can seamlessly integrate into MLLMs to improve the few-shot/zero-shot abilities on classification (upper right) and detection (bottom) datasets." This caption explicitly refers to "CLIP and MLLM", not "BERT". Image 2 (page 6) contains a figure caption that reads: "Fig. 2: Pipeline of RAR. (a) We design a multimodal retriever... (b)... we retrieve top-k categories from the memory and use MLLMs to refine the retrieved results as the final prediction through ranking." This caption explicitly refers to "RAR" and its pipeline. The user's question asks about the "pipeline diagram of the BERT model". The document images do not contain this information. They show figures for "CLIP and MLLM" and "RAR". The user's question is a hallucination trap. It asks for information about a model ("BERT") that is not mentioned in the document, while also referencing a specific diagram ("pipeline diagram") that is present but incorrectly attributed to a different model. Therefore, the question is not answerable because it is based on a false premise about the content of the document.

</think>

<answer>

**Not answerable**

</answer>

---

Figure 20: The model located the two most relevant pipeline diagrams in the document (Fig. 1 and Fig. 2). By analyzing their captions and content, it correctly determined that neither of them depicts the BERT model, thus identifying the user's question as unanswerable based on the provided text.

A.10  CASE STUDY

A.10.1  CASE STUDY: NEGATION BLINDNESS VS. SYSTEMATIC VERIFICATION

This example highlights a common failure mode in complex query understanding: **negation blindness**. The question requires the model to identify an option that is *not* present in the text, a task that demands more than simple keyword matching.

A base model, lacking a structured reasoning plan, tends to exhibit this failure. It is driven by information retrieval, successfully locating a passage that confirms one of the options (D) and incorrectly presenting it as the answer, thereby failing to address the negative constraint of the query.

In contrast, our CoR model employs a **systematic verification strategy**. It correctly interprets the task as a process of elimination and methodically checks each option against the source document. This robust process allows it to ignore misleading positive matches and correctly identify the truly absent option.

Figure 21 provides a side-by-side comparison of the final outputs, visually demonstrating the base model's failure and the success of our CoR-driven approach.

**Question:** *"Which of the following is NOT mentioned as a consideration for whaling-related CG operations?*
*A. Guidance for Cutters/Aviation, including D17 MMPA Guidance/D17INST/OPLAN/NEPA.*
*B. The dangers of whaling, particularly its occurrence ¿40NM offshore villages during whaling season.*
*C. The potential impact of commercial fishing vessels on whale populations.*
*D. The potential impact of research vessels on whales, including the risk of diverting westward migration.*
*Choose the letter name in front of the right option from A, B, C, D."*

### A.10.2  CASE STUDY: FAILURE IN NUMERICAL AND COMPARATIVE REASONING

This case study demonstrates a failure in fine-grained numerical and comparative reasoning, a critical task in document analysis. Answering the question requires not only locating the correct data but also accurately interpreting and comparing the values.

The base model successfully locates the correct evidence source (Figure 7) and even extracts the relevant numerical data for both low-wage (-17.7%) and high-wage (-16.9%) workers. However, it fails at the final, crucial reasoning step. It incorrectly concludes that a -16.9% decline is a more severe negative impact than a -17.7% decline, leading to a completely erroneous final answer.

This highlights a key advantage of our CoR model. Its explicit verification and reasoning steps would force a direct numerical comparison, correctly identifying that -17.7% represents a larger magnitude of decline. The stark difference in the models' final outputs is illustrated in Figure 22.

**Question:** *"In terms of employment rates, who is more negatively impacted by 'low-wage workers' vs 'high-wage workers'?"*

## B    STATEMENT ON THE USE OF LARGE LANGUAGE MODELS (LLMS)

During the preparation of this manuscript, the Large Language Models (LLMs) Grok and Gemini were utilized to assist with the literature review, as well as for language polishing and grammatical correction to improve clarity and readability. It is important to state that the role of these models was strictly supportive; they were not involved in the core conceptualization, experimental design, or data analysis of this research. The authors completed all academic work independently and assume full responsibility for the final content, ensuring it adheres to all scholarly standards.

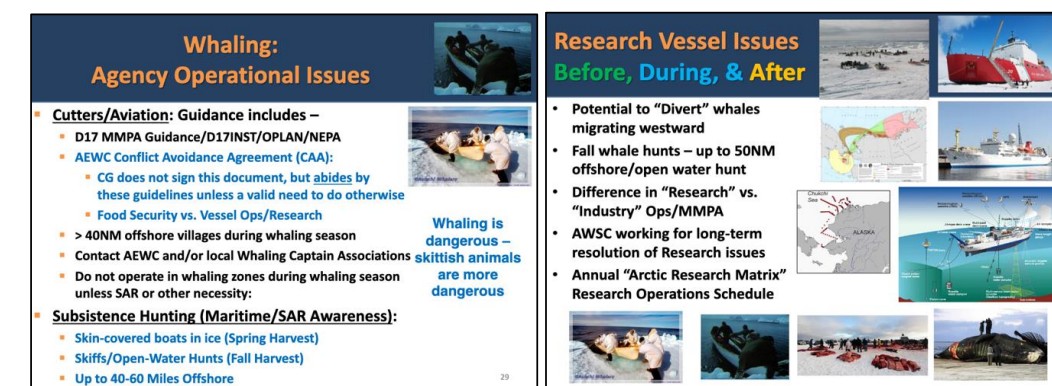

*Pages 29, 32 of 4034595.pdf (52 pages) in LongDocURL*

**Question:** Which of the following is NOT mentioned as a consideration for whaling-related CG operations?
A. Guidance for Cutters/Aviation, including D17 MMPA Guidance/D17INST/OPLAN/NEPA.
B. The dangers of whaling, particularly its occurrence >40NM offshore villages during whaling season.
C. The potential impact of commercial fishing vessels on whale populations.
D. The potential impact of research vessels on whales, including the risk of diverting westward migration.
Choose the letter name in front of the right option from A, B, C, D.
**Answer:** "C"

**Qwen2.5-VL** ❌
After reviewing the document's section on "Whaling: Agency Operational Issues," the text confirms that the potential impact of research vessels on whales, including the risk of diverting westward migration, is a mentioned consideration for CG operations. So the answer is D.

**Qwen2.5-VL-CoR-7B** ✅
\<think>
The user wants me to identify which of four options is *not* mentioned in the document. This requires me to verify the existence of each option and then report the one that is absent. First, I need to locate the relevant section. I'll search for keywords like \"whaling\" and \"CG operations\". I've found a section titled \"Whaling: Agency Operational Issues\". This seems like the most relevant place to find information about considerations for CG operations. Now, I will examine each of the four options presented in the question. Option A: \"Guidance for Cutters/Aviation, including D17 MMPA Guidance/D17INST/OPLAN/NEPA.\" This is explicitly listed under the \"Cutters/Aviation\" heading. Option B: \"The dangers of whaling...\". This is mentioned under the \"Subsistence Hunting (Maritime/SAR Awareness)\" heading. Option C: \"The potential impact of commercial fishing vessels on whale populations.\" I have searched the entire document, including the \"Whaling\" section, and there is no mention of commercial fishing vessels. Option D: \"The potential impact of research vessels on whales...\". This is mentioned under the \"Research Vessel Issues\" heading. Since option C is the only one that is not mentioned anywhere in the document, it is the correct answer.
\</think>
\<answer>
C
\</answer>

Figure 21: A side-by-side comparison for a negation-based question. The Base Model incorrectly matches keywords for option D and outputs it as the answer. Our CoR model correctly identifies the unmentioned option C through its systematic verification process.

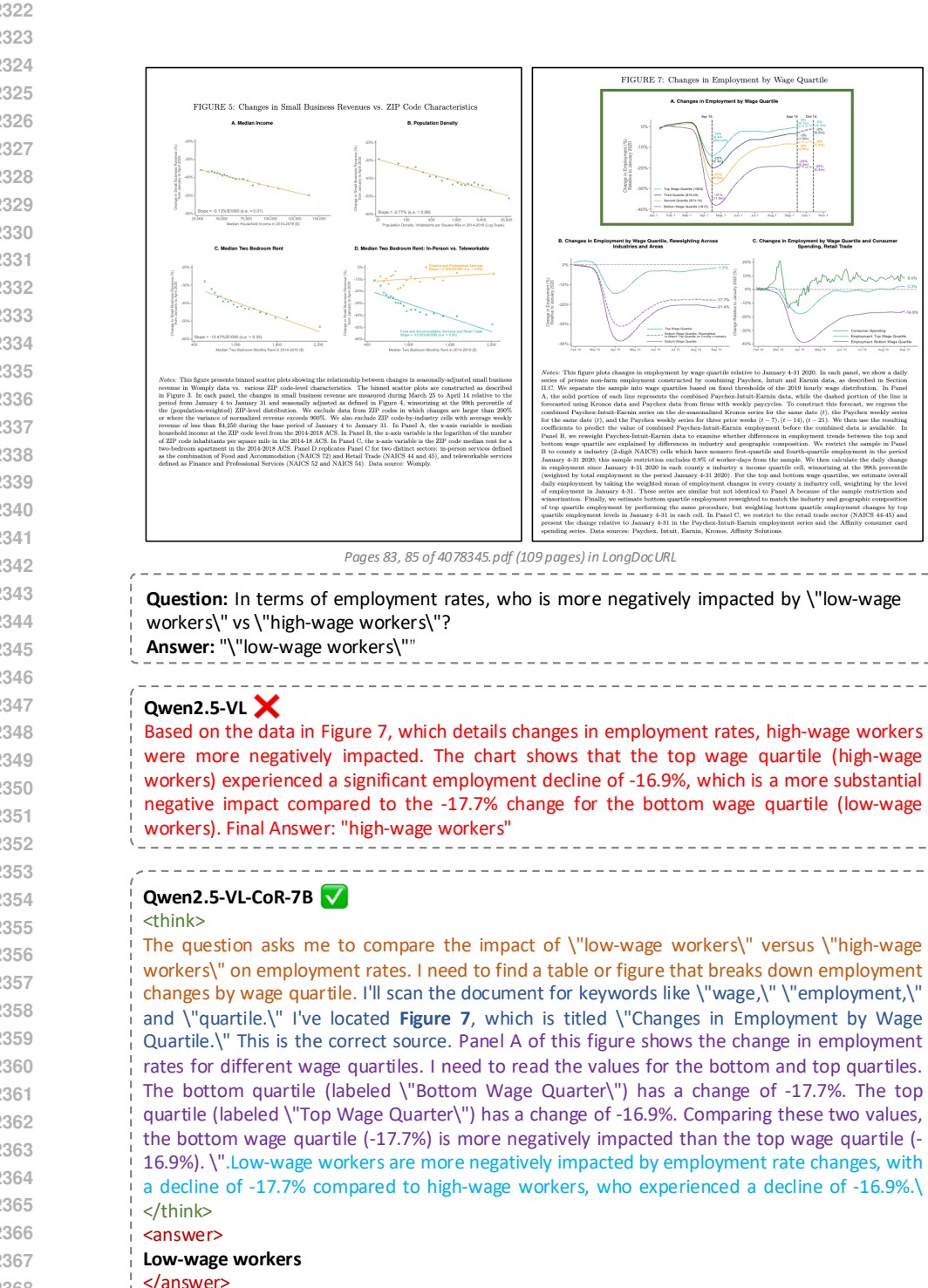

*Pages 83, 85 of 4078345.pdf (109 pages) in LongDocURL*

**Question:** In terms of employment rates, who is more negatively impacted by \"low-wage workers\" vs \"high-wage workers\"?
**Answer:** "\"low-wage workers\""

**Qwen2.5-VL** ❌
Based on the data in Figure 7, which details changes in employment rates, high-wage workers were more negatively impacted. The chart shows that the top wage quartile (high-wage workers) experienced a significant employment decline of -16.9%, which is a more substantial negative impact compared to the -17.7% change for the bottom wage quartile (low-wage workers). Final Answer: "high-wage workers"

**Qwen2.5-VL-CoR-7B** ✅
<think>
The question asks me to compare the impact of \"low-wage workers\" versus \"high-wage workers\" on employment rates. I need to find a table or figure that breaks down employment changes by wage quartile. I'll scan the document for keywords like \"wage,\" \"employment,\" and \"quartile.\" I've located **Figure 7**, which is titled \"Changes in Employment by Wage Quartile.\" This is the correct source. Panel A of this figure shows the change in employment rates for different wage quartiles. I need to read the values for the bottom and top quartiles. The bottom quartile (labeled \"Bottom Wage Quarter\") has a change of -17.7%. The top quartile (labeled \"Top Wage Quarter\") has a change of -16.9%. Comparing these two values, the bottom wage quartile (-17.7%) is more negatively impacted than the top wage quartile (-16.9%). \".Low-wage workers are more negatively impacted by employment rate changes, with a decline of -17.7% compared to high-wage workers, who experienced a decline of -16.9%.\
</think>
<answer>
**Low-wage workers**
</answer>

Figure 22: Figure 7 from the source document, showing changes in employment by wage quartile. Correctly answering the question depends on accurately comparing the values for the top and bottom quartiles.

