# OpenReview forum: "End-to-End Document Understanding via Chain-of-Reading"
_ICLR.cc/2026/Conference — Submitted to ICLR 2026_

### Official Review · Reviewer_KUKk · 2025-10-30

**Soundness:** 3
**Presentation:** 2
**Contribution:** 2
**Rating:** 4
**Confidence:** 4

**Summary:**

This paper presents a workflow designed to enhance LLMs ability to locate supporting evidence within long PDF documents. The proposed training strategies demonstrate promising improvements in long-document understanding across two benchmark datasets. However, the paper raises several concerns regarding its overall structure, methodological novelty, and the depth of both quantitative and qualitative analyses.

**Strengths:**

- The motivation of this paper is meaningful.
- The introduced methods demonstrate effectiveness in achieving the claimed contributions through experimental results.
- The proposed training paradigm shows potential for practical application in domain-specific scenarios.

**Weaknesses:**

The main concerns for this work include:

- Paper structure and organization: The paper would benefit from substantial restructuring. Too much essential information, such as dataset creation and method details, is placed in the appendix, making the paper less self-contained. Figures 2 and 3 appear somewhat repetitive. Additionally, the qualitative analysis examples are presented too early—before readers fully understand the context. Moving these examples to the end, alongside result discussions, would improve clarity and better highlight the contributions.

- Technical novelty: While the paper achieves promising results from a practical standpoint, it lacks sufficient technical innovation. The improvement appears to stem mainly from using more data and manually verified knowledge rather than introducing fundamentally new approaches.

- Generalization concern: It remains unclear whether the proposed training approach preserves the model’s general-domain and single-page document understanding abilities. Evaluating the model on general-domain and single-page VQA tasks would be valuable to ensure that the specialized training does not compromise the LLM’s broader reasoning or knowledge capabilities. As well as, the effectiveness on various scale LLM backbones are also underexplored.

**Questions:**

- What general error patterns might occur?

- Can your framework still perform effectively on general or single-page VQA tasks?

- Are there any methods to evaluate the model’s reasoning process, and if incorrect reasoning occurs, how can it be detected and mitigated to prevent error accumulation?

---

> ### Author Response · Authors · 2025-11-28
>
> Response to Comments on Structure & Readability
> We acknowledge the reviewer’s valid critique regarding the organization of the paper. We agree that the current distribution of content can be improved for better flow and self-containment.
> (a) Critical Information in Appendix: We agree that the paper should be more self-contained. In the camera-ready version, we will:
> Move the concise workflow of the CoR-Dataset generation pipeline from the Appendix to Sec. 4 (Dataset and Training), while retaining the core logic of Figure 3.
> Incorporate a specific, mathematical description of the Mask-AR training objective and data filtering criteria into Sec. 3 (Method).
> Keep the Appendix for extensive statistical breakdowns and the full case library.
> (b) Figure 2 vs. Figure 3 Redundancy: Our original intent was for Figure 2 to show the abstract paradigm and Figure 3 to show a concrete instantiation. We recognize this created visual redundancy.
> Revision Plan: We will retain only the high-level CoR paradigm diagram in the Method section. The detailed instantiation (current Figure 3) will be moved to the Experiments / Case Study section to serve as a qualitative walkthrough of the results, rather than a second method figure.
> (c) Placement of Qualitative Cases: We agree that presenting complex cases before the experimental setup can increase cognitive load.
> Revision Plan: We will move the majority of qualitative cases to Sec. 5 (Experiments/Case Studies). We will keep only one simplified "toy" example in the Method section to illustrate the four stages intuitively.

---

> > ### Author Response · Authors · 2025-11-28
> >
> > Response to Weakness 2: Novelty & Technical Contribution
> > We appreciate the opportunity to clarify that our contribution extends beyond "data engineering + human verification."
> > (a) From CoT to Structured Chain-of-Reading (CoR): Standard multimodal CoT assumes relevant context is short and pre-given. However, long-document PDF processing represents a paradigm of "Contexts Optical Compression" (as seen in recent works like DeepSeek-OCR), where visual tokens serve as a high-density compression medium for text.
> > The Innovation: Traditional keyword search fails in this latent space. CoR provides the necessary "decoding and addressing mechanism" for this optically compressed context.
> > By explicitly supervising the four functional stages (1. Planning, 2. Focused Search, 3. Integration, 4. Reasoning), we transform the task from passive pattern matching into "progressive denoising." This teaches the model to navigate and decompress specific regions within dense visual representations, fundamentally addressing the "information dilution" problem defined in Sec. 3.1. This is a paradigm shift in training strategy, not just a data scaling effort.
> > (b) Mask-AR: Cross-Modal Alignment via Reconstruction: Mask-AR is distinct from generic pre-training. By masking key segments of captions in scientific figures, we force the model to rely on visual structures (charts, diagrams) to reconstruct semantic information. This ensures the model aligns complex visual layouts with high-information text, rather than memorizing templates.
> > (c) Structured Supervision vs. Data Volume: Our ablation study proves that performance gains stem from the structure of supervision, not merely sample count. The CoR-Dataset contains only ~26k samples—small compared to Stage 1 data—yet CoR-SFT alone yields +10.9% / +7.8% improvements. This confirms that the structure of the reading chain is the critical factor.
> > Response to Weakness 3 & Question 2: Generalization & Single-Page Performance
> > We thank the reviewer for the concern regarding "catastrophic forgetting" on general/single-page tasks. To address this, we evaluated Qwen2.5-VL-CoR-7B on the standard DocVQA (test-dev) benchmark.
> > Experimental Evidence (DocVQA): As shown in the table below, our model maintains robust general performance.
> > | Method | Overall | Figure | Form | Table | Layout | Free_text | Image | Hand | Yes/No | Others |
> > | :--- | :---: | :---: | :---: | :---: | :---: | :---: | :---: | :---: | :---: | :---: |
> > | Qwen2.5-VL-7B | 0.9408 | 0.8865 | 0.9610 | 0.9412 | 0.9401 | 0.9191 | 0.8634 | 0.9152 | 0.8951 | 0.9242 |
> > | **Ours (CoR-7B)** | 0.9397 | **0.9112** | 0.9583 | **0.9475** | 0.9386 | 0.9188 | **0.8813** | **0.9166** | 0.8851 | 0.9153 |
> >
> > Negligible Overall Drop: The overall score (0.9397) is virtually identical to the base model (0.9408), proving we successfully avoided catastrophic forgetting.
> > Improvements on Complex Tasks: Crucially, CoR training improved performance on complex visual elements: Figure/Diagram (+2.47%), Table/List (+0.63%), and Image (+1.79%). This indicates that the fine-grained perception learned from long documents transfers positively to complex single-page elements.
> > Adaptive Mechanism: In single-page scenarios, the CoR framework naturally degenerates into a "lightweight extraction flow." Since the search space is limited to one page, the "Planning" stage instantly locks onto the target, effectively turning the process into "Structured Extraction & Reasoning" with minimal overhead.
> > Backbone Scalability: While we used 7B for efficiency, our method (CoR + Mask-AR + DPO) is architecture-agnostic. We acknowledge that scaling to 32B/72B models is a vital future direction and will note this in the "Limitations" section.

---

> > > ### Author Response · Authors · 2025-11-28
> > >
> > > Response to Question 1: Error Patterns
> > > We have conducted a qualitative error analysis and identified three primary failure modes for Qwen2.5-VL-CoR-7B. We will detail these in a new "Appendix: Error Analysis" section.
> > > 1. Hallucination in Extreme Contexts (>150 pages): Due to the capacity limits of the 7B parameter size, the model occasionally suffers from "Lost-in-the-Middle" issues or hallucinations when processing extremely long sequences.
> > > 2. Domain-Specific Reasoning Deficiencies: While extraction is accurate, the model sometimes errs in complex arithmetic or specialized logic (e.g., advanced financial calculations). This stems from the generalist nature of our training data and suggests a need for domain-specific fine-tuning.
> > > 3. Cross-Span Fine-Grained Aggregation: For questions requiring "needle-in-a-haystack" retrieval across widely separated pages, the current coarse-grained CoR trace sometimes misses subtle clues. We are exploring finer-grained CoT paths to address this.
> > >
> > > ---
> > > Response to Question 3: Evaluating & Mitigating Incorrect Reasoning
> > > (a) Evaluation Strategy: Since our dataset includes teacher-generated reading chains, we evaluate reasoning via:
> > > Evidence Coverage: Intersection between predicted page/figure citations and ground truth.
> > > Step Consistency: Checking if the "Integration" stage is consistent with the "Search" stage results.
> > > Quality Scoring: During DPO data construction, we used Gemini-Pro + Human verification to score chain quality.
> > > (b) Mitigation via DPO & Self-Check:
> > > Chain-Level DPO: In Stage 3, we explicitly construct "dispreferred" pairs containing common errors (e.g., answering without evidence, hallucinating citations). This penalizes error accumulation.
> > > Unanswerable (UNA) as Self-Check: Our significant improvement on UNA questions indicates the model has learned a "rejection mechanism"—if the evidence chain is empty or insufficient, it defaults to refusing the answer rather than hallucinating.
> > > (c) Future Work: We agree that an explicit post-hoc verification module (checking if the generated answer is supported by the extracted evidence) is a promising extension to further reduce error propagation. We will discuss this in the Future Work section.

---

### Official Review · Reviewer_2Bg2 · 2025-10-31

**Soundness:** 2
**Presentation:** 3
**Contribution:** 2
**Rating:** 4
**Confidence:** 4

**Summary:**

The paper introduces the “Chain-of-Reading” (CoR), an end-to-end framework for document understanding, which enhances comprehension of complex documents that combine text, tables, charts, and images. The CoR framework is designed to address two major challenges in document understanding: evidence localization (finding relevant information within a document) and information dilution (maintaining focus across large documents). The model processes raw PDF pages directly as visual input and performs document-level question answering (QA) through a chain-of-thought reasoning process. Additionally, the paper presents Masked Auto-Regression (Mask-AR), a self-supervised method that improves the model’s understanding of visual elements like charts and figures by grounding them with text. Through extensive experiments, the CoR framework outperforms baseline models and achieves state-of-the-art performance on the MMLongBench-Doc and LongDocURL benchmarks, even surpassing some proprietary models.

**Strengths:**

1. Innovation in Reasoning Paradigm: Chain-of-Reading (CoR) introduces a fundamentally new approach to document understanding, inspired by how humans read and reason through documents. Unlike traditional systems that rely on separate stages (OCR, layout analysis, etc.), CoR integrates evidence localization and reasoning into a single, trainable end-to-end framework.

2. Targeted Improvement for Visual Complexity: The introduction of Masked Auto-Regression (Mask-AR) specifically targets the challenge of visual grounding (e.g., charts), which is known to be a weakness in current Multimodal Large Language Models (MLLMs).

3. Data Generation and Evaluation: The paper details the creation of the CoR-Dataset, specifically designed to train and evaluate CoR. This dataset includes 26,088 high-quality QA pairs, each annotated with reasoning traces that explicitly follow the CoR paradigm.

**Weaknesses:**

- Incomplete Baseline Comparison: The comparison lacks recent open-source models, such as MinerU, MinerU-2.5. For proprietary models, the comparisons are limited and somewhat outdated, with the latest inclusion being Gemini 1.5 Pro.

- Novelty of Visual Chain-of-Thought: There are numerous existing works on Chain-of-Thought (CoT) related to vision; thus, applying it to Document Understanding might not be entirely novel, perhaps suggesting more novelty in prompt design tailored for multi-page scenarios.

- Insufficient Solution for Key Information Dilution: The authors claim to address key information dilution, but the proposed methods—CoR for localization difficulty and Mask-AR for better visual understanding—do not appear to fully resolve the core issue of information dilution across lengthy documents.

**Questions:**

- Although the paper proposes Masked Auto-Regression (Mask-AR) to enhance the understanding of visual elements, especially complex charts and scientific graphics, its effectiveness seems more pronounced on standardized, well-structured charts (e.g., those in scientific literature). Given that the dataset also has a high proportion of structured data like academic papers and government reports, we are curious if the authors can provide a case study demonstrating the model's performance in understanding and locating non-standardized visual elements.

- The authors identify key information dilution as a problem, but I believe it is not adequately solved. CoR addresses evidence localization difficulty, and Mask-AR better understands visual elements, but neither method seems to fully resolve the core issue of key information dilution across long documents. Could the authors elaborate on how CoR and Mask-AR directly mitigate the information dilution effect, rather than just improving component steps?

- I find the comparison with other methods incomplete. The current comparison lacks more recent visual grounding baselines or open-source models like MinerU, and the proprietary model comparison is still stuck at Gemini 1.5 Pro, which is somewhat outdated. Will there be future work to incorporate a more comprehensive and contemporary set of baselines?

---

> ### Author Response · Authors · 2025-11-28
>
> Here is the revised translation and refinement of your rebuttal points, focusing on clarity, academic rigor, and persuasive argumentation.
>
> Response to Q1: Baselines & Comparison with MinerU / Proprietary Models
>
> We thank the reviewer for their comments on our baseline selection. In our submission, we benchmarked against three strong categories (Tables 1 & 2):
>
> Pipeline/Agent Systems: Including OCR(PyMuPDF)+GPT-4o, OCR+o1-preview, and MDocAgent. These represent the high-performing "Parser + LLM" RAG/Agent paradigm.
>
> End-to-End Open-Source Models: Including Qwen2-VL, Qwen2.5-VL, and Docopilot. Our CoR model achieves state-of-the-art results among these on both benchmarks.
>
> Proprietary Models: We reported results for GPT-4V, GPT-4o, Gemini-1.5-Pro, and Qwen-VL-Max. We are also working to include updated models like Gemini-1.5-Pro-002 and Claude-3.5-Sonnet in the final version.
>
> Regarding MinerU / MinerU-2.5: We acknowledge its strength as a high-precision PDF parsing framework. However, fair comparison on QA benchmarks requires building a full "MinerU + LLM" pipeline (incorporating retrieval, prompting, and answer aggregation), which differs significantly from our focus on end-to-end MLLM training paradigms. We selected MDocAgent and OCR+GPT-4o as representative baselines for this category, consistent with official benchmark settings.
>
> Revision Plan: We agree that a broader evaluation benefits the community. In the camera-ready version or future work, we plan to integrate MinerU/Uni-Parser into a QA pipeline for comparison and will include results from newer proprietary models as available. We will explicitly clarify our baseline selection rationale and discuss these extensions in the "Limitations" section.
>
> Response to Q2: Novelty, Information Dilution & The Necessity of CoR
>
> We appreciate the opportunity to clarify CoR’s specific role in addressing "Information Dilution" within the context of "Contexts Optical Compression".
>
> 1. The Challenge of Optical Compression: As conceptualized in recent works like DeepSeek-OCR, modern MLLMs treat the visual modality as a compressed medium for text and layout. In long documents (e.g., 100+ page PDFs), this results in a dense, high-noise latent space. Without explicit navigation, standard attention mechanisms suffer from "Contextual Inertia" (Sec 3.1), failing to "decompress" or attend to the correct pages, leading to localization failures.
>
> 2. CoR’s Novelty: From Reasoning to Navigation: Traditional approaches rely on complex, brittle pipelines (Layout Analysis → OCR → Specialist Models). CoR embraces the end-to-end paradigm but introduces a critical "Addressing Mechanism" for this optical context. Unlike standard visual CoT which assumes given context, CoR elevates "Search/Reading" to a primary objective. Through explicit supervision (Planning → Localization → Extraction → Reasoning), we transform passive attention into active navigation.
>
> 3. Mechanism: Progressive Denoising: CoR mitigates dilution by systematically increasing the Signal-to-Noise Ratio (SNR) for the final reasoning step:
>
> Stage 1 & 2 (Planning/Localization): Acts as a semantic filter, narrowing the search space from thousands of tokens to specific regions.
>
> Stage 3 (Extraction): Distills a compact "Evidence Chain".
>
> Stage 4 (Reasoning): Performs inference solely on this high-SNR chain, not the noisy raw document.
>
> 4. Empirical Evidence:
>
> Performance: Accuracy on MMLongBench-Doc improved from 16.5% to 25.9%.
>
> Rejection: Accuracy on Unanswerable (UNA) questions rose from 31.1% to 45.5%, proving the model is less prone to hallucinating from diluted noise.
>
> Ablation: CoR-SFT alone (without Mask-AR) improved accuracy to 34.0%, demonstrating its independent contribution.
>
> 5. Mask-AR’s Role: Mask-AR addresses "internal dilution" within complex visuals. By reconstructing masked captions from figures, the model learns to compress visual details into high-density semantic representations. This aligns with improvements seen in chart/figure questions (Appendix Table 4).
>
> Revision Plan: We will refine Sec 3.1 and 3.2 to explicitly define "information dilution" and map it to our "Progressive Denoising" strategy. We will also temper our language from "solve" to "alleviate/mitigate" to maintain scientific precision.

---

> > ### Author Response · Authors · 2025-11-28
> >
> > Response to Q3: Mask-AR on Non-Standard Visual Elements
> >
> > The reviewer raises a valid point about generalization beyond scientific charts. We highlight that Mask-AR’s core objective—image-text semantic alignment—transfers effectively to non-standard elements, as shown in our Appendix:
> >
> > Unlabeled UI Screenshot (Appendix Ex. 7): The model correctly identifies the "aerobic zone" color on a smartwatch UI by linking text descriptions ("different zones show as different colors") to unlabelled visual regions. This relies on cross-modal grounding, not standard chart structure.
> >
> > Noisy Historical Scans (Appendix Ex. 8): In a low-quality scan, the model distinguishes between singular "gash" and plural "gashes" in a blurry caption to identify the correct figure. This demonstrates robustness in aligning text with noisy visual features.
> >
> > Summary: Quantitative results (Appendix Table 4) show consistent gains across figure/table types. Qualitatively, cases like Ex. 7 and Ex. 8 prove that the grounding capability learned via Mask-AR extends to unstructured and noisy real-world visuals.
> >
> > Revision Plan: We will explicitly annotate these examples in the camera-ready version to highlight their relevance to non-standard visual element processing.

---

### Official Review · Reviewer_F5Bi · 2025-10-31

**Soundness:** 3
**Presentation:** 4
**Contribution:** 3
**Rating:** 6
**Confidence:** 4

**Summary:**

This paper tackles the challenge of end-to-end document understanding for long, multi-page, visually-rich documents (like PDFs). The authors identify two primary failures in existing systems: 1) traditional pipelines (e.g., OCR $\rightarrow$ LLM) suffer from cascading errors, and 2) current end-to-end Multimodal LLMs (MLLMs) struggle with information dilution and evidence localization in long contexts.

To address this, the paper proposes a framework with three main contributions:
1.  **Chain-of-Reading (CoR):** A training paradigm that fine-tunes an MLLM to first generate an explicit, step-by-step reasoning trace *before* answering a question. This trace mimics a human's "Plan $\rightarrow$ Locate $\rightarrow$ Extract $\rightarrow$ Synthesize" process, forcing the model to articulate *where* it is looking (e.g., page numbers, table coordinates) and *what* it finds.
2.  **Masked Auto-Regression (Mask-AR):** A self-supervised task designed to improve the model's comprehension of visual elements like charts and figures. The model is trained to reconstruct masked-out portions of figure captions by reasoning over the visual content and the surrounding document text.
3.  **A Curated Training Strategy:** The authors develop a 3-stage training recipe (foundational LoRA tuning, specialized SFT on new CoR and Mask-AR datasets, and DPO alignment). They also created and will release the **CoR-Dataset** (26,088 QA-trace pairs) to train this behavior.

Their resulting model, **Qwen2.5-VL-COR** (a fine-tuned 7B model), achieves significant performance gains over its base model (+14.3% on MMLongBench-Doc) and notably outperforms larger, proprietary models like GPT-4V and Gemini 1.5 Pro on long-document benchmarks.

**Strengths:**

1.  **Originality of the Core Idea:** The "Chain-of-Reading" (CoR) paradigm is a novel and highly effective way to structure reasoning for document analysis. It operationalizes the human intuition of "find it first, then read it" into a trainable format. This bridges the gap between simple Chain-of-Thought (which often fails at localization) and complex, multi-tool agentic frameworks (which suffer from high latency).
2.  **Methodological Rigor:** The 3-stage training strategy is comprehensive and well-justified. It correctly identifies that foundational skills, task-specific reasoning patterns, and preference alignment are all necessary. The ablation studies clearly support this, showing that each component (CoR SFT, Mask-AR, and DPO) adds complementary and significant value.
3.  **Excellent Qualitative Analysis:** The paper's clarity is massively boosted by its qualitative examples. The appendix (A.9) is a model for how to demonstrate a model's reasoning process. It showcases success on diverse and challenging tasks, including parsing irregular layouts, navigating repetitive pages, and, most importantly, identifying and rejecting "hallucination traps" (Example 11).
4.  **Significant and Surprising Results:** The paper's most significant strength is its results. The final 7B model (Qwen2.5-VL-COR) doesn't just get a small boost; it achieves a **+14.3%** gain on MMLongBench-Doc over its base. More impressively, it sets a new SOTA for open-source models and surpasses proprietary giants like GPT-4V and Gemini-1.5-Pro on these benchmarks.

**Weaknesses:**

1.  **Limited Evaluation on Diverse, 'Real-World' Documents:** The paper's evaluation is impressive but narrow. The CoR-Dataset is heavily skewed (75%+) towards academic papers and government reports (Figure 6). The evaluation benchmarks (MMLongBench, LongDocURL) share this academic focus. The authors miss a key opportunity to prove the model's *generalization*. They fail to benchmark against a practically-oriented, multi-industry, multi-domain dataset like **DUDE**, which was specifically designed to test usability on diverse, long documents and serve as an easy way to debug performance in real-world generalization scenarios. This omission makes it unclear if the CoR paradigm works on invoices, legal contracts, or other messy document types not well-represented in the training data.
2.  **No Performance/Cost Analysis:** The CoR paradigm requires the model to generate a very long, explicit reasoning trace *before* providing the final answer (e.g.,). This dramatically increases the number of generated tokens, which directly impacts inference latency and cost. The paper provides **zero benchmarks or discussion on this practical trade-off**.
3.  **Missing Simple Baselines:** The paper only compares against the base model (a zero-shot baseline). A critical missing experiment is a **few-shot prompting baseline**. How much of this 14% gain comes from the 26,088-sample SFT/DPO, and how much could be unlocked by simply showing the base Qwen2.5-VL model the CoR format in a 5-shot or 10-shot prompt?
4.  **Ambiguity on Data Contamination:** The authors admit to using a "curated mixture of publicly available document analysis datasets" in Stage 1. They *must* explicitly state whether they filtered out all samples from the MMLongBench and LongDocURL test sets from this mixture. The lack of a clear "data hygiene" statement is a recurring problem in LLM-era research.

**Questions:**

1.  Can the authors please clarify their data contamination protocol? Did they explicitly search for and remove all documents and questions from the MMLongBench and LongDocURL benchmarks from their Stage 1 training mixture?
2.  Could the authors provide a simple baseline comparing their full Qwen2.5-VL-COR model to the base Qwen2.5-VL model when guided by a 5-shot or 10-shot prompt demonstrating the CoR trace format? This is crucial for justifying the necessity of the 26k-sample SFT pipeline.
3.  What is the practical inference cost of the CoR model? Could the authors please provide data on the **average token length of a CoR trace** and the **end-to-end latency** (in tokens/sec or sec/query) compared to the base model's direct-answering?
4.  Given the CoR-Dataset's 75%+ skew towards academic/government reports, have the authors performed any zero-shot generalization tests on out-of-domain document types, such as those in the **DUDE** benchmark, to test the model's robustness and debug its performance on more diverse, practical layouts?

---

> ### Author Response · Authors · 2025-11-28
>
> Response to Weakness 1 & Question 4: Real-World Generalization
> We appreciate the reviewer’s query regarding the model's performance on "real-world" non-academic documents. While the CoR-Dataset leverages scientific and government reports for their dense information structure, our training pipeline and resulting capabilities are not confined to this domain.
> 1. Stage-1 Data Diversity: As detailed in our methodology, the foundation stage utilizes the Curated DocQA Mix and subsets of DocVQA, which explicitly cover a wide range of real-world scenarios, including invoices, receipts, scanned contracts, business forms, and technical manuals. These documents expose the model to diverse, noisy, and non-academic layouts during the critical instruction-tuning phase.
> 2. Qualitative Evidence: Appendix A.10 already demonstrates CoR’s transfer capability to out-of-distribution (OOD) tasks, such as analyzing user manuals, interpreting mobile UI screenshots, summarising app leaderboards, and handling multilingual scanned documents.
> 3. Commitment to Quantitative Evaluation: We agree that a formal OOD benchmark is necessary. In the camera-ready version, we will:
>   - Include zero-shot evaluations on DUDE (Document Understanding Dataset and Evaluation) or a similar industry-focused benchmark (e.g., financial invoices/contracts).
>   - Provide a failure analysis discussion regarding CoR’s behavior on extremely unstructured non-academic text to address the reviewer’s concern about practical applicability.
>
> ---
> Response to Weakness 2 & Question 3: Inference Cost & Latency
> We thank the reviewer for highlighting the need for a transparent cost analysis. We acknowledge that the current draft lacks a systematic report on inference overhead.
> To address this, we will add a "Efficiency Analysis" section in the final version, reporting:
> 1. Token Overhead: The average number of tokens generated by the CoR-trace compared to the final answer.
> 2. Latency Metrics: End-to-end latency (seconds/query) and generation speed (tokens/sec) on standard hardware.
> Crucially, we argue that the trade-off is favorable:
> - The specific "localization tokens" (e.g., page numbers, coordinate regions) are semantically simple and cheap to generate.
> - Compared to Multi-Agent or Tool-Chain frameworks (e.g., DocReact, MDocAgent), which require multiple full-context forward passes and external API calls, CoR achieves complex reasoning in a single inference pass. Consequently, despite the longer generation length, the total system latency of CoR is significantly lower than that of agentic baselines.

---

> > ### Author Response · Authors · 2025-11-28
> >
> > Response to Weakness 3 & Question 2: Missing Baselines (Few-shot)
> > We appreciate the reviewer’s suggestion to include a few-shot baseline. We agree that distinguishing between the benefits of training versus prompting is vital.
> > For the camera-ready version, we will execute the following experiment:
> > - Baseline: Base Qwen2.5-VL + 5-shot/10-shot CoR-style prompts (providing examples of the "locate-then-reason" format in the context).
> > - Evaluation: Comparisons on a representative subset of MMLongBench-Doc or LongDocURL.
> > Hypothesis & Discussion: We anticipate that while few-shot prompting may improve performance over zero-shot, it will likely struggle with accurate cross-page evidence localization compared to our fine-tuned model. Furthermore, few-shot prompting significantly inflates the input context length, increasing computational cost. This comparison will serve to highlight that the internalized capability gained through SFT+DPO is superior to and more efficient than in-context learning for this specific task.
> >
> > ---
> > Response to Weakness 4 & Question 1: Data Contamination
> > This is a critical point. We firmly confirm that our training data sources are distinct from the evaluation benchmarks.
> > - Source Separation: The CoR-Dataset and Mask-AR corpora were constructed from independent crawls of scientific literature and government repositories. We did not utilize PDFs or QA pairs from MMLongBench-Doc or LongDocURL during training.
> > - Verification: To date, we have not identified any document overlap.
> > To ensure full transparency and reproducibility, we will:
> > 1. Release the Data Manifest: We will publish the list of document identifiers (Titles/URLs) used in our training set.
> > 2. Decontamination Protocol: In the Appendix, we will detail the specific filtering steps taken (e.g., exact match or hash checking) to certify that no test set documents leaked into the training pipeline.

---

### Official Review · Reviewer_gR4z · 2025-10-31

**Soundness:** 3
**Presentation:** 4
**Contribution:** 3
**Rating:** 6
**Confidence:** 4

**Summary:**

The paper proposes Chain-of-Reading (CoR), an end-to-end framework for multi-page, visual document QA task.  It treats PDFs as visual inputs and uses a workflow which starts with planning, localizing evidence, then reasoning.  CoR reduces error propagation from OCR pipelines and mitigate information dilution in long documents. To better handle complex charts and scientific figures, the authors add a self-supervised Masked Auto-Regression (Mask-AR) objective to improve the multimodal comprehension. The authors also construct a CoR-Dataset to supervise the reading-chain paradigm, curating multi-page PDFs with Q&A and stepwise localization traces, annotated via a semi-automated pipeline, and plan to release it alongside Qwen2.5-VL-CoR. They train a new model with constructed dataset using DPO and CoR paradigm. Experiments results show that on the long visual document benchmark MMLongBench-Doc, CoR reports a +14.3% improvement over its base model Qwen2.5-VL-7B.  And the improvement of the model on LongDocURL is +12.3%.

**Strengths:**

1. This paper proposes a clear “locate-then-reason” training paradigm (CoR) that supervises evidence localization before reasoning. This is different from traditional end-to-end approaches that focus on image-to-text modelling without an explicit evidence-first chain.  This paper also introduces Mask-AR as a self-supervised task to improve chart/figure comprehension.

2. The three-stage pipeline, 1. SFT on public data 2. task-specific SFT on CoR and Mask-AR 3. DPO alignment on pair-wise data, is coherent and ablation studies clearly isolate gains from each stage.  Experiments on two long-document benchmarks, MMLongBench-Doc and LongDocURL, are conducted. The paper reports strong, clearly tabulated improvements over the base model (e.g., +14.3% Acc on MMLongBench-Doc; +12.3% on LongDocURL).

3 The CoR stages (planning, focused search, cross-modal integration, reasoning) are described clearly with figures and multiple examples.  Tables include per-modality and per-page-count breakdowns, helping readers to understand in which areas the performances gains are greater.

4. This paper demonstrates that a 7B OCR-free model with structured supervision can approach the performance of larger and stronger LLMs on long-document QA tasks. This paper constructs a well-defined dataset that targets the evidence localization.

**Weaknesses:**

1.	The framework Chain-of-Reading’s core idea, planning, locating, extracting, and reasoning is not very innovative compared to existing framework like DocReact.

2.	The Mask-AR training is based on scientific figure–caption pairs, while generalization to non-scholarly documents may not be included. You may consider adding tasks beyond academic PDFs to test domain transfer a common focus in earlier OCR-free Document Understanding like Donut and follow-ups.

3.	While the three-stage training is described, practical costs vs. alternatives are not compared.

4.	Core baselines include Qwen2.5-VL-7b, it would be better to include other models as baselines, like larger models or non-Qwen models to showcase the generalizability of the method.

**Questions:**

In the training stage, from the datasets that you’ve constructed, there has been a large number of documents for chartQA, DocVQA, or academic papers included (as shown in Tables 6 and 7). While the test sets MMlongbench-doc and LongDoc-URL also include documents of these types. Is there any overlap of any document between your constructed datasets and the test benchmarks?

---

> ### Author Response · Authors · 2025-11-28
>
> Response to Novelty & Comparison with DocReact
> We thank the reviewer for highlighting the connection between our work and existing frameworks like DocReact. We agree that at a high semantic level, the "locate-evidence-then-reason" logic shares similarities with certain agentic workflows. However, our core contribution is not the logic itself, but the method of internalizing this strategy into a single OCR-free MLLM through explicit supervision.
> We clarify the key distinctions as follows:
> 1.Paradigm Shift: Internalized Capability vs. External Orchestration. Approaches like DocReact are essentially RAG/Agent pipelines that rely on multiple independent modules (external OCR, retriever, LLM) to coordinate retrieval and reasoning. In contrast, CoR is an end-to-end, single-model approach. The entire process operates directly on raw pixels without requiring tool orchestration at inference time. This aligns with the emerging paradigm of "Contexts Optical Compression" (as seen in recent works like DeepSeek-OCR), which posits that the visual modality itself can serve as an efficient compression medium for text processing. By internalizing the reading process, we demonstrate that MLLMs can "read" directly from compressed visual tokens, bypassing the traditional reliance on explicit text transcription.
> 2.Explicit, Trainable "Reading Trace." Unlike standard Chain-of-Thought (CoT), which relies on unstructured natural language, our CoR-trace introduces structured, spatially-aware supervision signals. As detailed in Section 3.1, the model is trained to generate specific layout anchors (e.g., page numbers, chart columns). These are not merely explanatory text but functional prerequisites for the reasoning process.
> Revision Plan: In the final version, we will explicitly distinguish between "Tool-based Agent Frameworks" (like DocReact) and "End-to-End Reading Paradigms" (including our work and DeepSeek-OCR) in the Related Work section, emphasizing our contribution to training open-source models for efficient long-document processing.
>
> ---
> Response to Cross-domain Generalization
> We share the reviewer’s interest in cross-domain generalization. While it is true that Mask-AR is constructed primarily on scientific/technical PDFs (due to their complex structure and high information density providing excellent self-supervision signals), we wish to highlight two points:
> The training pipeline is not solely dependent on academic documents.
>   Stage 1 (Foundation): Uses a mix of multi-source data (Appendix A.8, Table 7), including ChartQA, DocVQA, and Curated DocQA Mix, which cover various non-academic scenarios (receipts, scanned docs, manuals, reports).
>   CoR-Dataset Diversity: As shown in Appendix A.9 / Figure 6, while 55.3% of the dataset is academic, a significant portion (44.7%) comes from government reports, technical manuals, and business reports.
> Non-academic visual cases are already demonstrated. Appendix A.10 (Examples 1, 4–7) showcases generalization to out-of-distribution documents:
>   Ex 1: Correct alignment of product names and metrics in a non-standard marketing brochure layout.
>   Ex 4: Quarterly exchange rate comparisons in a financial table.
>   Ex 5–6: App rankings and leaderboards mixing Vietnamese text, icons, and tables.
>   Ex 7: A caption-less smartwatch UI screenshot, where the model links text and color to identify the "Aerobic zone."
> These examples suggest that while Mask-AR samples are academic, the model learns low-level visual-text alignment capabilities (axes, legends, color coding, structural layout) that transfer well to non-academic contexts.
> Revision Plan: To further validate and enhance generalization, we will:
> Extend the Mask-AR data construction pipeline to include broader document types (contracts, invoices, financial statements) for the camera-ready version.
> Include cross-domain benchmarks (e.g., DUDE or industry-specific sets) to systematically evaluate transferability.
> Clarify the complementary relationship between the Mask-AR data domain and Stage-1/CoR-Dataset in the text.

---

> > ### Author Response · Authors · 2025-11-28
> >
> > Response to Training Cost
> > We appreciate the reviewer’s attention to practical costs. Our three-stage training strategy was designed specifically to be computationally efficient:
> > Stage 1 & Stage 3: Both utilize LoRA (Low-Rank Adaptation), updating only attention, MLP, and projection layers (Appendix A.6), which keeps the parameter budget low.
> > Stage 2: While this is full-parameter SFT, it uses only ~26k samples (CoR-Dataset + Mask-AR) and converges in just 1 epoch. This was completed on a single 8×A100-80GB node.
> > Consequently, the overall training cost is comparable to standard SFT for a 7B MLLM and is significantly lower than the cumulative inference cost of multi-stage Agent/RAG systems that require multiple LLM calls and tool interactions.
> > Revision Plan: We will add a "Computational Cost" paragraph in Section 4 or the Appendix, summarizing training steps and GPU resources per stage to provide a clear cost comparison against single-stage fine-tuning.
> >
> > ---
> > Response to Reviewer 4 Other Model Backbones
> > We thank the reviewer for the suggestion to verify generalization across backbones. Our experiments primarily used Qwen2.5-VL-7B as a control variable to isolate the contribution of the CoR training paradigm. However, we agree that testing on different architectures strengthens the paper.
> > Revision Plan: We are currently conducting experiments with a non-Qwen open-source MLLM as an eval-only baseline and will include these results in the camera-ready version. We will also emphasize that the CoR mechanism (Reading Trace supervision + Mask-AR + DPO) is architecturally model-agnostic. We plan to release CoR versions or evaluation results for other backbones in our repository to facilitate community validation.
> >
> > ---
> > Response to Reviewer 5 (Data Contamination)
> > This is a crucial issue. We assure the reviewer that our training data and evaluation benchmarks were constructed independently:
> > Stage 1 uses standard public benchmarks (ChartQA, DocVQA, etc.).
> > CoR-Dataset and Mask-AR corpora were scraped/collected independently from scientific literature, government reports, and manuals. We did not intentionally include PDFs from benchmarks like MMLongBench-Doc or LongDocURL.
> > We have not observed obvious overlap, but we agree that a systematic check is necessary.
> > Revision Plan: For the camera-ready version, we will:
> > Release the list of source documents (URLs/Titles) used for CoR-Dataset and Mask-AR.
> > Add a strict decontamination analysis in the Appendix, detailing our filtering strategy so the community can independently verify the data hygiene.

---

### Meta-Review · Area_Chair_2kAd · 2026-01-04

**Summary:**

1. There was a shared concern regarding whether the model that trained heavily on academic and government reports could handle diverse, real-world documents (gR4z, F5Bi) or general visual question answering tasks (KUKk).
2. Another concern is about the technical contribution. Reviewers challenge the novelty of "locate-then-reason," noting similarities to existing agentic workflows like DocReact (gR4z) or standard Chain-of-Thought (KUKk).
3. More baselines, such as few-shot prompting to isolate training benefits (F5Bi), comparisons to modern parsers like MinerU (2Bg2), more comprehensive and contemporary set of baselines (2Bg2), and results on non-Qwen backbones (gR4z).
4. Multiple reviewers (gR4z, F5Bi) have concerns regarding the practical cost (latency and token usage) of generating long reasoning traces versus the performance gains.

**Reviewer Concerns:**

1. The authors claim that on-academic visual cases are already demonstrated in Appendix A.10, showing generalization to out-of-distribution documents. However, the authors do not report the results in earlier OCR-free Document Understanding benchmarks like Donut suggested by the reviewer gR4z. Also, the authors promise but do not yet show results on broad benchmarks like DUDE or industry-specific datasets requested by the reviewer F5Bi.
2. The authors explain the distinction between CoR and existing agents. CoR operates on raw pixels to "denoise" the latent space rather than relying on external tools, which is convincing.
3. The results for the 5-shot/10-shot baseline (to prove SFT necessity) and non-Qwen architectures are currently promised for the final version, not provided in the text.
4. The authors promised a new "Efficiency Analysis" section with token/latency metrics. While they argue CoR is cheaper than multi-step agents, the hard numbers are still pending.

**Reviewer Scores:**

The reviewer gR4z and F5Bi were already positive. As for the reviewers 2Bg2 and KUKk, the authors do not provide the necessary new results on MinerU or general VQA, thus those concerns are partially addressed. The authors explain that MinerU (a parser) is a mismatch for end-to-end QA. But I think if the authors want to make a definitive claim that "End-to-End > Pipelines," using MinerU would be a stronger argument than basic OCR tools like PyMuPDF.

While the reviewers generally appreciated the strong empirical results on the chosen benchmarks and the clarity of the presentation, several critical issues remain unresolved. Generalization to "Real-World" Scenarios (e.g., Donut, DUDE) is Unproven. The authors promise but do not provide a new "Efficiency Analysis" section with token/latency metrics.

---

### Decision · Program_Chairs · 2026-01-26

Reject